# AUTOMATIC FUNCTIONAL DIFFERENTIATION IN JAX

**Min Lin**
Sea AI Lab
linmin@sea.com

## ABSTRACT

We extend JAX with the capability to automatically differentiate higher-order functions (functionals and operators). By representing functions as a generalization of arrays, we seamlessly use JAX's existing primitive system to implement higher-order functions. We present a set of primitive operators that serve as foundational building blocks for constructing several key types of functionals. For every introduced primitive operator, we derive and implement both linearization and transposition rules, aligning with JAX's internal protocols for forward and reverse mode automatic differentiation. This enhancement allows for functional differentiation in the same syntax traditionally use for functions. The resulting functional gradients are themselves functions ready to be invoked in python. We showcase this tool's efficacy and simplicity through applications where functional derivatives are indispensable. The source code of this work is released at https://github.com/sail-sg/autofd.

## 1 INTRODUCTION

In functional analysis, *functions* can be characterized as continuous generalizations of vectors. Correspondingly, the principles of linear algebra, derivatives, and calculus for vectors can be extended to the spaces of functions. Based on this generalization, we can build higher-order functions whose domain/codomain are function spaces. These higher-order functions are traditionally called *functionals* or *operators*. For example, definite integral is a functional that maps a function to a scalar, and the differential operator is an operator that maps a function to its derivative function. The generalization of gradient descent to functionals is particularly interesting to the machine learning audiences. Like we optimize a function by moving the input in the gradient direction, a functional can be optimized by varying the input function using the *functional derivative*. Gradient descent for functionals has been used for a long history in science and engineering, named the calculus of variations. Well known examples include the least action principle in Lagrangian mechanics, and the variational method in quantum mechanics.

While the theoretical framework for functional differentiation is established, there remains a notable gap in computational tools tailored to automatically compute functional derivatives. To this date, functional differentiation with respect to functions are usually done in one of the following ways: (1) manually derived by human and explicitly implemented (Zahariev et al., 2013; Gonis, 2014); (2) for semi-local functionals, convert it to the canonical Euler-Lagrange form, which can be implemented generally using existing AD tools (Hu et al., 2023; Cranmer et al., 2020). In stark contrast, the domain of automatic differentiation (AD) for functions that maps real tensors has seen significant advancement recently and widespread application in various domains, e.g. robotic simulation (Freeman et al., 2021), cosmology (Piras & Mancini, 2023) and molecular dynamics (Schoenholz & Cubuk, 2021). Like AD has transformed several fields and industries, we believe automatic functional differentiation (AutoFD) holds the potential to catalyze a similar wave of innovation.

Based on the idea that functions can be represented as a generalization of arrays, we propose that AutoFD can be implemented in the same vein as AD. Building AutoFD in this way avoids the complexity of symbolic approaches that depends on analytical integrals. JAX provides an elegant machinery for supporting both forward mode and reverse mode AD without redundant code. The forward mode differentiation is implemented via linearization rules, also called jacobian vector product (JVP) rules for each primitive computation. The reverse mode differentiation consists of three steps (Radul et al., 2023), 1. perform forward mode AD on a function; 2. unzip the linear and

nonlinear part of the function; 3. transposition of the linear part. Therefore, to make a primitive computation differentiable in JAX, it is crucial that the primitive is implemented with a *JVP rule* and a *transpose rule*. Similarly, AutoFD for an operator relies on higher-order generalizations of JVP and transpose rules, which are well defined in mathematics. The Fréchet derivative extends the idea of derivative to higher-order functions whose inputs and outputs are functions. And the adjoint operator generalizes the concept of transposition from function to higher order functions.

Having introduced the fundamental mechanisms for AutoFD, we now delve into the specific operators we include in this work. We motivate the choice of operators from the types of operators and functionals that are commonly used, which we summarize as non-linear semi-local functionals and linear non-local operators. These two types cover the most used functionals in various applications, for instance, the majority of exchange-correlation functionals in density functional theory fall under semi-local or linear non-local functional categories (Perdew & Schmidt, 2001). It turns out that to build any functional described in Section 2, there is a set of five essential operators, namely, compose, $\nabla$, linearize, linear transpose and integrate. These operators, along with their JVP and transpose rules, are detailed in Section 3.2. To ensure the completeness of the operator set, we also introduce some auxiliary operators in Section 3.3. We discuss the applications of AutoFD in Section 4, presenting both opportunities for new methods and improvements on the coding style.

## 2 OPERATORS AND FUNCTIONALS

By convention, we define operators as mappings whose domain and codomain are both functions. For example, the $\nabla$ operator when applied on a function returns the derivative of that function. Functionals are mappings from functions to scalars. To distinguish them from plain functions that map real values, we denote both functionals and operators with capital letters, with an extra *hat* for operators. We use round brackets to denote the application of operators on functions as well as the application of functions on real values. e.g. $\hat{O}(f)(x)$ means application of operator $\hat{O}$ on function $f$, and apply the resulting function on $x$. Without loss of generality, we write integrals without specifying the domain. For simplicity, we present the results on scalar functions when there is no ambiguity. The actual implementation supports functions that map arbitrarily shaped tensors or nested data structure of them. The operators we aim to support can be categorized into three different types based on their property.

**Local operator** generalizes element-wise transformation in finite dimensional vector spaces. Consider the input function $f : X \rightarrow Y$, with any function $\phi_L : X \times Y \rightarrow Z$, a local operator $\hat{\Phi}_L$ needs to satisfy

$$\hat{\Phi}_L(f) : X \rightarrow Z; x \mapsto \phi_L(x, f(x)). \tag{1}$$

**Semilocal operator** extends local operator by introducing extra dependencies on the derivatives of the input function up to a finite maximum order $n$.

$$\hat{\Phi}_S(f) : X \rightarrow Z; x \mapsto \phi_S(x, f(x), \nabla f(x), \nabla^{(2)} f(x), \cdots, \nabla^{(n)} f(x)). \tag{2}$$

This type of operator is called semilocal because $\hat{\Phi}_S(f)(x)$ not only depend on the value of $f(x)$ at the point $x$, but also on the function values in the infinitesimal neighborhood, i.e. $f(x + \delta x)$, via the knowledge of the derivatives.

**Nonlocal operators** are the operators that are neither local nor semilocal. Although they do not need to assume any specific forms, one of the most interesting nonlocal operator is the *integral transform*,

$$\hat{\Phi}_I(f) : U \rightarrow Y; u \mapsto \int \phi_I(u, x) f(x) dx. \tag{3}$$

The function $\phi_I : U \times X \rightarrow Y$ is called the *kernel function*. Integral transform generalizes finite dimensional matrix vector product; therefore it is a linear mapping of $f$. The Schwartz kernel theorem (Ehrenpreis, 1956) states that any linear operators can be expressed in this form.

**Integral functionals** are functionals that conform to the following form:

$$F : \mathcal{F} \rightarrow \mathbb{R}; f \mapsto \int \hat{\Phi}(f)(x) dx. \tag{4}$$

Correspondingly, integral functionals are called local, semilocal or nonlocal depending on the property of operator $\hat{\Phi}$. The components used to build functional approximators in existing works belong to one of the above types. For example, the fourier neural operator (Li et al., 2020) is a direct generalization of multilayer perceptron, the linear layers are nonlocal *integral transforms*, while the nonlinear activation layers are pointwise transformation that follows the form of local operator. The lagrangian neural networks (Cranmer et al., 2020) is implicitly a composition of a integral functional with a learnable semilocal operator. In applications like density functional theory, most empirically developed functionals belongs to the semi-local family. Recently, non-local functionals are also introduced for better treatments of electron correlation (Margraf & Reuter, 2021).

## 3 IMPLEMENTATION

### 3.1 GENERALIZED ARRAY

In JAX, array is a fundamental data structure that corresponds to vector, matrix and tensor in mathematics. Primitives are algebraic operations for transforming the arrays. Following the notation of JAX, we use `f[3,5]` to describe an float array of shape 3 by 5. To represent functions as generalied arrays, we first generalize the notations. The shape of a function is represented as a list of its return value and each arguments in the format of `F[ret,arg0,⋯]`. For example, `F[f[],f[3],f[2,3]]` describes a function: $\mathbb{R}^3 \times \mathbb{R}^{2\times3} \to \mathbb{R}$. In AutoFD, this abstract information is implemented as a custom class and it is registered via `jax.core.pytype_aval_mappings[types.FunctionType]` to make JAX recognize and convert python functions into generalized arrays when tracing higher order function.

### 3.2 PRIMITIVE OPERATORS

The primitive operators considered in this work are focused to realize the most used types of operators and functionals described in Section 2. To enable functional derivatives for these primitives, as mentioned in the introduction, we need to define a JVP rule and a tranpose rule for each of them. Here we only make connections between the programming notation and math notation of the JVP and tranpose rules, which should be enough for understanding the implementation of AutoFD. For formal definitions, we refer the readers to Section II.5 in Sternberg (1999) for tangent/cotangent vectors, and Definition 3.6.5 in Balakrishnan (2012) for Fréchet derivative.

The JVP function in JAX is defined as

$$\texttt{jax.jvp}(f, \underbrace{x}_{\text{primal}}, \underbrace{\delta x}_{\text{tangent}}) = J_f(x)\delta x = \underbrace{D(f)(x)(\delta x)}_{\text{Fréchet notation}}. \tag{5}$$

Where $J_f(x) = \frac{\partial f}{\partial x}\big|_x$ is the jacobian of function $f$ at point $x$. The JVP function is also called the forward mode gradient, it evaluates change of function $f$ at point $x$ along the direction $\delta x$. However, backward mode gradient is what we usually use in backpropagation. In JAX, the function associated with backward gradient is vector jacobian product (VJP), VJP means vector is on the left hand side of the jacobian, but we usually transpose it instead to obtain a column vector.

$$\texttt{jax.vjp}(f, \underbrace{x}_{\text{primal}})(\underbrace{\delta y}_{\text{cotangent}}) = J_f(x)^\top \delta y$$

Usually, forward mode and backward mode are implemented separately. However, Radul et al. (2023) introduces that idea that the VJP is simply the linear transposition of JVP. Here we introduce the notations of linear transpose in programming and in math. With $f(x) = Mx;\ f^\top(y) = M^\top y$,

$$\texttt{jax.linear\_transpose}(f, \underbrace{x}_{\text{primal}})(\underbrace{y}_{\text{cotangent}}) = f^\top(y) = \underbrace{T(f)(x)(y)}_{\text{our own Fréchet like notation}} \tag{6}$$

It can be easily seen that when we apply linear transposition to JVP: $\delta x \mapsto J_f(x)\delta x$, it results in the VJP: $\delta y \mapsto J_f(x)^\top \delta y$. Therefore, in JAX, each primitive is given a JVP rule and a transpose rule, together they are used to derive the backward mode gradient in JAX.

The JVP and transpose rules can be generalized to higher order functions. We use the Fréchet derivatives $D$ and $\partial_i$ to denote JVP rules and partial JVP rules. $D(\hat{O})(f)(\delta f)$ denotes the JVP of

operator $\hat{O}$ with the primal value $f$ and tangent value $\delta f$. We add a $\delta$ prefix for functions from the tangent and cotangent space. Similarly to the Fréchet notation, we introduce the symbol $T$ and $T_i$ to denote transpose and partial transpose rules. Note that although transpose rules generally do not require a primal value, partial transposes do sometimes need the primal values when the operator is not jointly linear on the inputs. Therefore, we explicitly write out the primal values for transpose rules for consistency; i.e. $T_f(\hat{O})(f, g)(\delta h)$ denotes transposition of the operator $\hat{O}$ with respect to $f$, with the primal values $f$, $g$ and cotangent value $\delta h$. The generalization of transposition to operators is the adjoint of the operator $T(\hat{O}) = \hat{O}^*$, satisfying $\langle \hat{O}f, g \rangle = \langle f, \hat{O}^*g \rangle$.

To support high order derivative, the JVP and transpose rules need to be implemented using primitives exclusively, which means any operator used in the right hand side of the JVP and transpose rules needs to be implemented as a primitive themselves. For example, the JVP rule (7) of the compose operator uses the linearize operator $\hat{L}$, which is described in Section 3.2.3. To save space, we leave the proof of these rules to the Appendix.

### 3.2.1 THE COMPOSE OPERATOR

Function composition is a frequently used operator, with $g : X \to Y; f : Y \to Z$. The compose operator is defined as $f \circ g : X \to Z; x \mapsto f(g(x))$. Here the compose operator $\circ$ is written as an infix operator, alternatively, we can use the symbol $\hat{C}$ to represent the compose operator, and $\hat{C}(f, g)$ to denote the composition of $f$ and $g$. The compose operator can be generalized to more than two functions. For example, $\hat{C}(f, g_1, g_2)$ describes the function $x \mapsto f(g_1(x), g_2(x))$. Implementation of compose in python is simple

```
def compose_impl(f, g):
  return lambda *args: f(g(*args))
```

The JVP and transpose rules of compose are derived as follows:

$$\partial_g(\hat{C})(f, g) : \delta g \mapsto \hat{C}(\hat{L}(f), g, \delta g). \tag{7}$$

$$\partial_f(\hat{C})(f, g) : \delta f \mapsto \hat{C}(\delta f, g). \tag{8}$$

$$T_f(\hat{C})(f, g) : \begin{cases} \delta h \mapsto \hat{C}(\delta h, g^{-1})|\det \nabla(g^{-1})| & \text{if } g \text{ is invertible.} \\ \texttt{undefined} & \text{otherwise.} \end{cases} \tag{9}$$

$$T_g(\hat{C})(f, g) : \begin{cases} \delta h \mapsto \hat{C}(\hat{T}(f), \delta h) & \text{if } f \text{ is linear.} \\ \texttt{undefined} & \text{otherwise.} \end{cases} \tag{10}$$

As can be seen, the implementation of the compose operator requires two auxiliary operators $\hat{L}$(linearize) and $\hat{T}$(transpose), which will be defined independently in Section 3.2.3 and 3.2.4. The function inverse on the right hand side of the $T_f(\hat{C})$ rule is not implemented because currently a general mechanism for checking the invertibility and for defining inversion rules of the primitives are not yet available in JAX. The compose operator is the basis for all local operators defined in (1). Replacing the dependency on $x$ with an identity function $I$, all local operators can be composed with $\hat{C}(\phi_L, I, f) : x \mapsto \phi_L(I(x), f(x))$. A large number of common operators can be defined via $\hat{C}$, for example, we overload the infix operators in python to support the syntax $f + g$ which converts to $\hat{C}((x, y) \mapsto x + y, f, g)$. All direct algebraic operations on functions in our code examples are supported in a similar way.

### 3.2.2 THE $\nabla$ (NABLA) OPERATOR

The $\nabla$ operator converts a function to its derivative function. In JAX, the corresponding transformations are `jax.jacfwd` and `jax.jacrev`. Its JVP and transpose rules are:

$$D(\nabla)(f) : \delta f \mapsto \nabla(\delta f). \tag{11}$$

$$T(\nabla)(f) : \delta h \mapsto -\nabla \cdot \delta h. \tag{12}$$

The implementation of the $\nabla$ operator is readily available in JAX as `jax.jacfwd` and `jax.jacrev`. The JVP rule is the same as the primal computation because $\nabla$ is a linear operator. The transpose

rule uses $\nabla$ as the *divergence* operator, which can be implemented as $\hat{C}(\text{trace}, \nabla(\delta h))$. We provide a quick proof of the transpose rule here for scalar variable functions, the full derivation is provided in Appendix C.2. To see the transpose rule, we write the $\nabla$ operator in the Schwartz kernel form, i.e. $\nabla = \int dy\ \delta'(x-y)$. Like finite dimensional linear transposition swaps the axis that is contracted, the $T(\nabla) = \int dx\ \delta'(x-y)$ simply changes the variable under integration, using integral by parts, we can verify that

$$T(\nabla)(f)(y) = \int dx\ \delta'(x-y)f(x) = \delta(x-y)f(x)|_{-\infty}^{+\infty} - \int dy\ \delta(x-y)f'(x) = -\nabla(f)(y).$$

The $\nabla$ operator can be used for constructing semilocal operators together with the $\hat{C}$ operator, i.e. semilocal operator in the form of (2) can be implemented as $\hat{C}(\phi_S, I, f, \nabla f, \cdots, \nabla^{(n)} f)$, where $I : x \mapsto x$ is the identity function, and $\nabla^{(n)}$ is the repeated application of $\nabla$ for $n$ times.

### 3.2.3 THE LINEARIZE OPERATOR

The linearize operator has the same meaning as the Fréchet derivative $D$. Except that $D$ can be used with any higher-order functions, while we reserve the symbol $\hat{L}(f) : x, \delta x \mapsto \nabla(f)(x)\delta x$ for the linearization of functions. The linearize operator is necessary for the completeness of local operators, as it is used in the JVP rule (7) of the compose operator. The JVP and transpose rules of the linearize operator are defined as following:

$$D(\hat{L})(f) : \delta f \mapsto \hat{L}(\delta f). \tag{13}$$

$$T(\hat{L})(f) : \delta h \mapsto -\nabla \hat{I}_y(y\delta h(\cdot, y)). \tag{14}$$

The $\hat{I}_y$ symbol used in the transpose rule represents integration over the variable $y$. The corresponding function transformation implemented in JAX is `jax.jvp`.

### 3.2.4 THE LINEAR TRANSPOSE OPERATOR

The linear transpose operator can be applied only on linear functions. It is easy to see that the operator itself is also linear. We only manage to derive a general form for the adjoint of the transpose operator when the cotangent function is invertible.

$$D(\hat{T})(f) : \delta f \mapsto \hat{T}(\delta f). \tag{15}$$

$$T(\hat{T})(f) : \begin{cases} \delta h \mapsto \delta h^{-1} |\det \nabla \delta h^{-1}| & \text{if } \delta h \text{ is invertible.} \\ \texttt{undefined} & \text{otherwise.} \end{cases} \tag{16}$$

The corresponding transformation defined in JAX for linear transposition is `jax.linear_transpose`.

### 3.2.5 THE INTEGRATE FUNCTIONAL

The integrate functional indispensable component for building functionals. It is required in the integral functionals (4), for constructing nonlocal operators (3), and used in the transpose rule (14) of the linearize operator. The integrate functional is called an integrate operator when we perform integration over only part of the variables. We use $\hat{I}$ for integrate operator, with a subscript for the variable under integration. For a function of two variables, $\hat{I}_x(f) : f \mapsto \int f(x,y)dx$. The JVP rule and transpose rules are:

$$D(\hat{I}_x)(f) : \delta f \mapsto \hat{I}_x(\delta f) \tag{17}$$

$$T(\hat{I}_x)(f) : \delta h \mapsto (x, y \mapsto \delta h(y)) \tag{18}$$

Only a limited subset of functions admit analytical integrals, the Risch algorithm provides a robust mechanism for determining the existence of an elementary integral and for computing it. In practical applications, numerical integration techniques are more commonly employed. These techniques are rooted in the concept that a function can be expanded in a series of integrable functions. Methods of numerical integration often hinge on a designated grid of integration points associated with respective weights. For example, the Fourier series converts to summation over a uniform grid, and the Gaussian quadrature induces points that are the roots of the Legendre/Laguerre polynomials. With the grids and corresponding weights, integration of a function converts to a weighted summation over a finite set of points. We implement the integrate operator by supplying a numerical grid for the integrand function.

### 3.3 COMPLETENESS OF THE PRIMITIVES

Besides the above listed operators, a few other operators are needed for completeness. For example, the operator to permute the order that arguments are passed to a function: $\text{PermuteArgs}_{21}(\text{f})$ : $x, y \mapsto f(y, x)$, the operator to zip together two functions: $\text{Zip}(f, g) : x, y \mapsto f(x), g(y)$. We can inspect whether these rules forms a closed set by examining whether the right hand side of the rules (7) to (18) can be implemented with these primitives themselves. Most of the rules are readily implementable with the primitives, with a few exceptions:

1. Rule (9) and rule (16) requires the inverse of a function, which is not implemented because the general mechanism to invert a function is not available in JAX. When the function is a non-invertible map, the transpose rules can still exist mathematically (Equation 25). However, it has to be implemented case by case based on specific properties of the mapping.
2. Rule (14) involves an integral, as discussed in Section 3.2.5, there is no general rule for integration for any functions. Therefore, the accuracy of the integral is limited by the chosen numerical grid.

### 3.4 EFFICIENCY OF THE IMPLEMENTATION

There are a few practical issues considered in our system. Firstly, chain of function composition is expensive, if we have a function $h$, and we compose with the expression $f \circ h + g \circ h$. In python code this is `lambda x: f(h(x)) + g(h(x))`, without proper treatment, `h(x)` will be invoked twice. The situation is exacerbated when the resulting function is again composed with other functions multiple times. Although in JAX, common sub-expression elimination (CSE) is performed to remove redundant computations during the JIT compilation, we find in practice that the computation graph could grow to a prohibitively large size before we could JIT it. To resolve this problem, we add a cache for the function calls. When the same function is invoked several times with the same argument, the cached output is directly returned. With a large amount of redundant computations, this optimization greatly reduced the computation time of the resulting function (Appendix D).

Another performance issue hides in the $\nabla$ operator. For one, the reverse mode derivative approximately doubles the nodes in the computation graph, the size of the graph grows quickly when we perform higher order derivatives. Moreover, it is quite common in applications that we need to perform various orders of differentiation on different variables, e.g. $\nabla_y(\nabla_x(f))(x, y)$ and $\nabla_y(f)(x, y)$. It is worth noting that the inner differentiation of the first expression $\nabla_x(f)$ shares computation graph with the second expression. Again, it could be prohibitively expensive for the tracing phase. This issue is not specific to our system but in general exists in JAX. We can see an opportunity for optimization using our system because we build a graph made of $\nabla$ operators and functions without actually executing it. It is desirable to compile mixed order differentiation of functions into efficient programs, for example, Faà di Bruno mode differentiation (Bettencourt et al., 2019). We leave this optimization as a future work.

## 4 APPLICATIONS

In numerous application scenarios, function approximation problems are often transformed into parameter fitting tasks, such as using neural networks or linear combinations of function bases. Gradients in the function space may appear unnecessary, as our ultimate optimization takes place in the parameter space. In this section, we focus on application cases where functional derivative plays a crucial role.

### 4.1 SOLVING VARIATIONAL PROBLEM

Variational problems are problems that search for the optima of a functional. We consider local functional of the form $F(y) = \int I(y^\theta(x), \nabla y^\theta(x))dx$. Conceptually, we just need to compute the functional derivative $\frac{\delta F}{\delta y}$ and perform gradient descent for functionals. In practice, to make it implementable, we usually introduce a parametric function $y^\theta(x)$. For optimization, we can directly compute the gradient w.r.t. $\theta$,

$$\frac{\partial}{\partial \theta} F(y^\theta) = \int \frac{\partial}{\partial \theta} I(y^\theta(x), \nabla y^\theta(x))dx \tag{19}$$

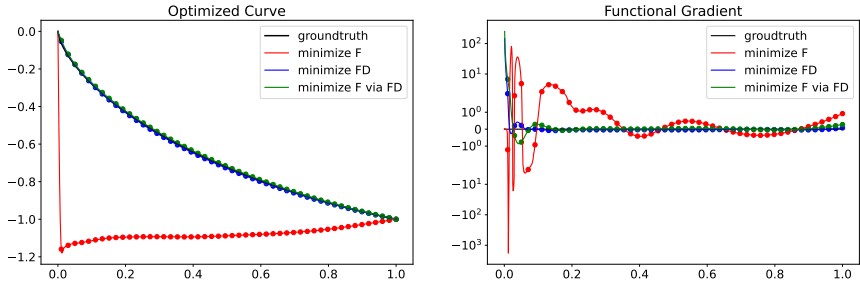

Figure 1: Brachistochrone curve fitted with different loss

While the above is the most straightforward method for this problem, with the introduction of functional gradient, we enable a few more options. One of the new options is to perform chain rule through the functional gradient, namely, we compute the functional derivative $\frac{\delta T(y)}{\delta y}$ first, then pointwisely backpropagate the gradient to the parameter,

$$\frac{\partial}{\partial \theta} F(y^\theta) = \int \frac{\delta T(y^\theta)}{\delta y^\theta}(x) \frac{\partial y^\theta(x)}{\partial \theta} dx \qquad (20)$$

Notice that in Equation (20) the functional derivative can be explicitly expanded using the Euler-Lagrange formula. With some derivation (Appendix E), we can see that Equation (20) and (19) are two different estimators for the gradient of $\theta$, though they share the same expectation.

One more choice we have is to directly minimize the magnitude of the functional gradient based on the fact that at the optimal, functional gradient should be zero.

$$\min_\theta \int \|\frac{\delta T(y^\theta)}{\delta y^\theta}(x)\|^2 dx \qquad (21)$$

We apply the above methods on the classic *brachistochrone* problem. The brachistochrone curve represents the optimal path that minimizes the travel time for a particle moving between two points in the gravitational field. Assume the initial position $x, y = 0, 0$, and the end position $x, y = 1, -1$. The travel time can be represented as a functional of the curve, denoted as $F(y) = \int_0^1 \sqrt{1 + \nabla y(x)^2} / \sqrt{-y(x)} dx$. The numerator calculates the length of the curve, the denominator is the speed of particle. We use $y^\theta(x) = \text{nn}^\theta(x) \sin(\pi x) - x$ as the approximator, where $\text{nn}^\theta$ is an MLP and the other terms are used to constrain the initial and end positions. More details of the experimental settings can be found in Appendix G

Instead of implementing Equation (20) explicitly, we illustrated below the simplicity of using AutoFD to compose the functional and compute functional gradient. The results are summarized in Figure 1, the methods (19), (20) and (21) are named *minimize F*, *minimize F via FD* and *minimize FD* correspondingly. It is worth highlighting that the directly minimize in the parameter space is limited by the numerical integration, it easily overfits to the numerical grid. Meanwhile, the functional derivative involved in the other two methods seems to provide a strong regularization effect, leading to curves close to the groundtruth. Better functional optimal are found as can be seen in Figure 1 (right) that the functional gradient is closer to zero. Whether Equation (20) is better than Equation (19) in a general context needs further investigation.

```python
from autofd.operators import integrate, sqrt, nabla

def y(params, x):
  ...

def F(y: Callable) -> Callable:
  return integrate(sqrt(1+nabla(y, argnums=1)**2)/sqrt(-y))
fd = jax.grad(F)(y)
```

## 4.2 DENSITY FUNCTIONAL THEORY

Density functional theory (DFT) is widely used for calculating electronic structures. Although the DFT theory proves the existence of a functional that maps electron density to the ground state energy, the accuracy of DFT in practice relies heavily on approximated exchange-correlation (XC) energy functionals with the general form

$$E_{\mathrm{xc}}(\rho) = \int \rho(\boldsymbol{r})\epsilon_{\mathrm{xc}}(\rho)(\boldsymbol{r})dr \tag{22}$$

Where $\rho$ is the electron density function, and $\epsilon_{xc}(\rho)$ is the XC energy density which is dependent on $\rho$. A comprehensive list of XC functionals are maintained in LibXC (Lehtola et al., 2018), its JAX translation is introduced in JAX-XC (Zheng & Lin, 2023). In the Self-Consistent Field (SCF) update of the DFT calculation, the effective potential induced by the XC functional is computed as its functional derivative $V_{\mathrm{xc}} = \frac{\delta E_{\mathrm{xc}}(\rho)}{\delta\rho}$. Traditionally, the $V_{\mathrm{xc}}$ is derived via the Euler-Lagrange equation. Its implementation is often done in the parameter space of $\rho$ and entangled with the numerical integral scheme. With automatic functional differentiation, the computation of $V_{\mathrm{xc}}$ can be decoupled from the choice of parameterization and integral schemes. In the code example below, gga_x_pbe is implemented as an semilocal operator using the primitives defined in this work. $V_{\mathrm{xc}}$ can be obtained by calling jax.grad on $E_{\mathrm{xc}}$.

```python
from autofd.operators import integrate
# a point in 3D
r = jnp.array([0.1, 0.2, 0.3])
# Exc, exchange-correlation energy functional
def exc(rho: Callable) -> Float:
  epsilon_xc = gga_x_pbe(rho)
  return integrate(epsilon_xc * rho)
# Vxc is the functional derivative of Exc
vxc = jax.grad(exc)(rho)
# Vxc is itself a function callable with r as input
vxc(r)
```

## 4.3 DIFFERENTIATING NONLOCAL FUNCTIONALS

Both the brachistochrone and DFT shown in previous sections are examples of integral functionals with semi-local functionals that conforms to 2. In this section, we explore a more interesting case where a nonlocal operator is involved. We consider the higher order version of multilayer perceptron (MLP), where the input is a $\mathbb{R} \to \mathbb{R}$ function. The $i$th linear layers follows $f_i(x) = \int k(x,y)f_{i-1}(y)dy + b(x)$, where $k$ is the kernel function resembling the weights in MLP, and $b$ acts as the bias. Activation functions are applied pointwise after the linear layer. For the last layer, we don't apply any activation, but compares the the output with a target function $t$ with $L_2$ loss. This is often called neural operator or neural functional, or which the Fourier Neural Operator (FNO) is a special case where the integral is done via fourier series.

To learn a neural operator, traditionally we use parameterized neural networks as $k^\theta(x,y)$ and $b^\phi(x)$ where $\theta, \phi$ are network parameters. The gradient descent is directly done in the parameter space, it seems like we can't apply AutoFD here. There is, however, an expensive way to do it, we can directly subtract the functional gradient from the old function, i.e. $k \leftarrow k - \eta\frac{\delta L(k)}{\delta k}$, where $\eta$ is the learning rate and both $k$ and $\frac{\delta L(k)}{\delta k}$ are functions. It is expensive because the new $k$ expands in the computation graph. The purpose of this experiment is only to show that AutoFD is capable of handling such complex computations. We show in Appendix H that our proposed update moves in the direction of smaller losses, though each step gets more expensive and yields more complex $k$.

## 5 RELATED WORKS

AD has a long history of wide applications across various domains in science and engineering. More recently, the success of deep learning has brought automatic differentiation to a focal point in machine

learning. In the world of machine learning, several differentiable programming frameworks backed by highly optimized vectorized implementations are proposed in the past few years, for example, Theano (Bastien et al., 2012), Mxnet (Chen et al., 2015), Tensorflow (Abadi et al., 2016), Jax (Frostig et al., 2018) and Torch (Paszke et al., 2019). All of these frameworks focus on AD of functions that map real values. Differentiation of functionals and operators, however, has been mostly derived analytically and implemented case by case. There are a number of works that studies AD in the context of higher-order functionals (Pearlmutter & Siskind, 2008; Elliott, 2018; Shaikhha et al., 2019; Wang et al., 2019; Huot et al., 2020; Sherman et al., 2021). They mainly focus on how AD can be implemented more efficiently and in a compositional manner when the code contains high-order functional transformations, which is related to but distinct from this work. For example, Elliott (2018) studies how $D(f \circ g)(x)$ can be implemented more efficiently using the algebraic relation of $D(f \circ g)(x) = D(f)(g(x)) \circ D(g)(x)$; while in this work, we study the differentiation of the higher-order function $\circ$ itself, namely $D(\circ)(f)$.

The most closely related works are Birkisson & Driscoll (2012) and Di Gianantonio et al. (2022). Birkisson & Driscoll (2012) computes the Fréchet derivative of operators by representing functions as linear combination of Chebyshev basis functions. Di Gianantonio et al. (2022) describes a language called Dual PCF which is capable of evaluating forward mode directional derivatives of functionals. The method is based on dual numbers and there is no reverse mode support mentioned. There are also implementations of functional derivatives in symbolic math packages. For example, Both Mathmematica (Wolfram Research) and SymPy (Meurer et al., 2017) implement the semi-local functional derivative in the canonical Euler-Lagrange form. In Maple (Maplesoft), there is also the FunDiff command, which relies on the calculus of Dirac delta function and its derivatives based on the provided information. Their implementation differs from the AD approach we take in this work, and they do not support vectorized execution on modern hardwares.

## 6 DISCUSSION

In this work, we introduce AutoFD, a system to perform automatic functional differentiation. We take a novel approach to functional differentiation by directly reusing the AD machinery for higher order functions. We implemented a core set of operators that covers various useful types of functionals. We discuss several limitations here as potential directions for future work.

**Completeness**: As discussed in Section 3.3, in several of the rules, the inversion operator is required. It would rely on a systematic mechanism to register the invertibility and inverse function for the primitives, at the time of writing, such mechanism is not implemented in JAX.

**Analytical integration**: It is desirable in applications like quantum chemistry to use analytical functions and integrate them analytically. While integrating symbolic packages like SymPy to JAX (Kidger) could provide this functionality, it is limited to scalar functions. Automatically de-vmapping the vectorized primitive to scalar functions could be one potential path to generally bring analytical integral to JAX.

**Static shape**: AutoFD requires accurate annotation of functions using `jaxtyping`. This is a design choice to allow early raising of errors as it is more informative than delayed to the execution of resulting functions. However, this not only adds extra work but also limits the flexibility of using AutoFD. Further exploration is required for a better trade-off.

**Programming in mixed order**: For example, the `partial` transformation in python is a mixed order operator that binds an argument to a function (both considered inputs to the partial operator). While it is possible to support gradients for both the argument and the function. Complications emerge during the just in time compilation, jitting a mixed computation graph is not possible because the operator primitives are pure python and do not support lowering. Ultimately we would like to remove this constraint and program differentiably for real values as well as any order of functions.

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

## A    EXAMPLE OF IMPLEMENTATION

To clarify how the math are correspondingly implemented as extension to JAX, we show a minimal implementation of the operator $\nabla$. We restrict the implementation to take only scalar function, so that the divergence $\nabla \cdot \delta h$ is equal to the gradient $\nabla \delta h$. With this simplification, the JVP and transpose rules are

$$D(\nabla)(f) : \delta f \mapsto \nabla(\delta f). \tag{23}$$

$$T(\nabla)(f) : \delta h \mapsto -\nabla \delta h. \tag{24}$$

Here's a list of mappings between math symbols and the code.

- $f$: `f`
- $\delta f$: `df`
- $\delta h$: `dh`
- $\nabla$: `nabla`
- $D(\nabla)(f)(\delta f)$: `nabla_jvp_rule((f,), (df,))`
- $T(\nabla)(f)(\delta h)$: `nabla_transpose_rule(dh, f)`

We first implement $\nabla$ as a JAX primitive.

```python
nabla_p = core.Primitive("nabla")

def nabla(f):
  return nabla_p.bind(f)

@nabla_p.def_impl
def nabla_impl(f):
  return jax.grad(f)

@nabla_p.def_abstract_eval
def nabla_abstract_eval(f):
  # f has scalar input and output
  # jax.grad(f) has same signature as f
  return f.shape

def nabla_jvp_rule(primals, tangents):
  # nabla is a linear operator
  f, df = primals[0], tangents[0]
  return nabla(f), nabla(df)

def nabla_transpose_rule(cotangent, primal):
  # According to the transpose rule in math
  dh = cotangent
  # we assume here negation on a function
  # is already implemented by the compose operator
  return -nabla(cotangent)

# we register the jvp and transpose rules.
jax.interpreters.ad.primitive_jvps[nabla_p] = nabla_jvp_rule
jax.interpreters.ad.primitive_transposes[nabla_p] = nabla_transpose_rule
```

Now we define some random functions as primal, tangent and cotangent values.

```python
def f(x):
    return jnp.sin(x)

def df(x):
    return x ** 2

def dh(x):
    return jnp.exp(x)
```

Finally, we show how the operator can be invoked, and how to perform automatic functional differentiation on the operator.

```python
# We can use nabla directly on f,
nf = nabla(f)
# triggers nabla_impl, nf: jnp.cos

# Or, we can compute the jvp of nabla, remember nabla is an operator
# jax.jvp here is computing the forward mode gradient for an operator!
nf, ndf = jax.jvp(nabla, primals=(f,), tangents=(df,))
# triggers nabla_jvp_rule, nf: jnp.cos, ndf: lambda x: 2x

# Since nabla is an linear operator, we can transpose it.
tnabla = jax.linear_transpose(nabla, primals=(f,))
# linear transpose of an operator is still an operator,
# we apply this new operator tnabla on the function dh.
tndh = tnabla(dh)
# triggers nabla_transpose_rule, tndh: lambda x: -jnp.exp(x)

# Or, we can do the backward mode gradient on nabla
primal_out, vjp_function = jax.vjp(nabla, f)
# invoke the vjp function on the cotangent dh
vjp_dh = vjp_function(dh)
# this triggers both nabla_jvp_rule and nabla_transpose_rule
# vjp_dh: lambda x: -jnp.exp(x)
```

# B  PROOFS OF JVP RULES

JVP rules are trivial for linear operator, for a linear operator $\hat{O}$, the JVP rule are always simply applying the same operator on the tangent function,

$$D(\hat{O})(f) : \delta f \mapsto \hat{O}(\delta f)$$

For our core set of operators, $\nabla$, $\hat{L}$ and $\hat{T}$ are all linear operators that need no extra proof for the JVP rules. We give here a step by step derivation for the JVP rules of the $\hat{C}$ operator.

$$
\begin{aligned}
\partial_g(\hat{C})(f, g)(\delta g)(x) &= \lim_{\tau \to 0} \frac{f(g(x) + \tau \delta g(x)) - f(g(x))}{\tau} \\
&= \frac{d}{d\tau} f(g(x) + \tau \delta g(x)) \mid_{\tau=0} = \nabla(f)(g(x)) \cdot \delta g(x) \\
&= \hat{C}(\hat{L}(f), g, \delta g)(x)
\end{aligned}
$$

$$\partial_f(\hat{C})(f,g)(\delta f)(x) = \lim_{\tau \to 0} \frac{f(g(x)) + \tau \delta f(g(x)) - f(g(x))}{\tau}$$
$$= \delta f(g(x))$$
$$= \hat{C}(\delta f, g)(x)$$

## C  PROOFS OF TRANSPOSE RULES

Given an operator $\hat{O}$ the adjoint of an operator $\hat{O}^*$ satisfies $(\hat{O}u, v) = (u, \hat{O}^*v)$.

### C.1  COMPOSE

The Schwartz kernel form of compose operator is $\hat{C}(f,g) = \int \delta(x - g(y))f(x)dx$.

$$\langle \hat{C}(f,g), \delta h \rangle = \int dy \delta h(y) \int dx \delta(x - g(y))f(x)$$
$$= \int dx f(x) \int dy \delta(x - g(y))\delta h(y)$$
$$= \langle f, \int dy \delta(x - g(y))\delta h(y) \rangle \qquad (25)$$
$$= \int dx f(x) \int dz |\det \nabla(g^{-1})(z)| \delta(x - z)\delta h(g^{-1}(z))$$
$$= \int dx f(x) |\det \nabla(g^{-1})(x)| \delta h(g^{-1}(x))$$
$$= \langle f, \hat{C}(\delta h, g^{-1}) | \det \nabla(g^{-1})| \rangle$$

Therefore, transposition of $\hat{C}$ w.r.t. $f$ is,

$$T_f(\hat{C})(f,g)(\delta h) = \hat{C}(\delta h, g^{-1}) |\det \nabla(g^{-1})|.$$

In the case where $f$ is linear, we can write $f(x)$ as $J_f x$ where $J_f$ is the jacobian matrix of $f$. We omit the case where $f$ is nonlinear, as transposition is only defined and implemented for linear operators. Transposition w.r.t. $g$ is simple

$$\langle \hat{C}(f,g), \delta h \rangle = \int (Fg(y))^\top \delta h(y) dy$$
$$= \int g^\top(y) F^\top \delta h(y) dy$$
$$= \langle g, \hat{C}(\hat{T}(f), \delta h) \rangle$$

Therefore, transpose of $\hat{C}$ w.r.t. $g$ is,

$$T_g(\hat{C})(f,g)(\delta h) = \hat{C}(\hat{T}(f), \delta h)$$

### C.2  NABLA

A brief proof for single variable function was given in the main text, here we expand to the multivariate case.

$$\langle \nabla f, g \rangle = \sum_{ij} \int dy g_{ij}(y) \int dx \delta'(x_j - y_j) f_i(y_{\sim j}, x_j)$$
$$= \sum_{ij} \int dy \delta'(x_j - y_j) g_{ij}(y) \int f_i(y_{\sim j}, x_j) dx$$
$$= \langle f, -\nabla \cdot g \rangle$$

## C.3 LINEARIZE

$$\langle \hat{L}(f), \delta h \rangle = \iiint dz \delta'(z - x) f(z) \delta x \cdot \delta h(x, \delta x) dx d\delta x$$

$$= \int dz f(z) \cdot \int d\delta x \; \delta x \int dx \delta'(z - x) \delta h(x, \delta x)$$

$$= \int dz f(z) \cdot \left( - \int \delta x \nabla \delta h(\cdot, \delta x) d\delta x \right)$$

$$= \langle f, - \int \delta x \nabla \delta h(\cdot, \delta x) d\delta x \rangle$$

Therefore,

$$T(\hat{L})(f)(\delta h) = - \int \delta x \nabla \delta h(\cdot, \delta x) d\delta x$$

## C.4 LINEAR TRANSPOSE

Linear transpose of linear transpose operator sounds interesting. Since linear transpose can only be applied to linear functions, we write the function being transposed as $x \mapsto J_f x$, transpose of $T(f) : y \mapsto J_f^\top y$. We're only able to derive the adjoint of $\hat{T}$ when $\delta h$ is an invertible mapping.

$$\langle \hat{T}(f), \delta h \rangle = \int (J_f^\top y) \cdot \delta h(y) dy$$

$$= \int y^\top J_f \delta h(y) dy$$

$$= \int \delta h^{-1}(x)^\top J_f x d\delta h^{-1}(x)$$

$$= \int |\det \nabla (\delta h^{-1})(x)| \delta h^{-1}(x)^\top J_f x dx$$

$$= \langle f, |\det \nabla (\delta h^{-1})| \delta h^{-1} \rangle$$

Therefore,

$$T(\hat{T})(f)(\delta h) = \hat{T}^*(\delta h) = |\det \nabla (\delta h^{-1})| \delta h^{-1}$$

## C.5 INTEGRAL OPERATOR

$$\langle \hat{I}_i(f), \delta h \rangle = \int dx_{\sim i} \int dx_i f(x_i, x_{\sim i}) \delta h(x_{\sim i})$$

$$= \int f(x) \bar{\delta} h(x) dx = \langle f, \bar{\delta} h \rangle$$

Where $\bar{\delta} h$ is $x_i, x_{\sim i} \mapsto \delta h(x_{\sim i})$. Therefore the adjoint of the integral operator is simply augmenting the cotangent function $\delta h$ with unused arguments to match the domain of the $f$.

# D COMMON SUBEXPRESSION ELIMINATION VIA CACHING

In Section 3.4, we introduced the composition $f \circ h + g \circ h$. We can recursively nest this composition by setting $h_i = f \circ h_{i-1} + g \circ h_{i-1}$. The redundant computations are then exponential to the depth of the composition. Specifically, we use the following code for measuring the execution cost for different nested depth, with and without function call caching.

```python
# we use sin, exp, tanh for h, f, g respectively.
def F(h):
  for _ in range(depth):
    h = f(h) + g(h)
  return h

# Fh is h = f(h) + g(h) nested to depth times
Fh = F(h)
# time the execution
t1 = time.time()
jax.jit(Fh)(0.)
t2 = time.time()
cost = t2 - t1
```

When the above nested function composition are implemented naively, the time cost of computation grows exponentially with the nest depth, because there are two branches f(h) and g(h) at each composition. However, when function call caching is used, at each level one branch can reuse the cached result of the other branch, resulting in a linear time cost with respect to the nest depth. We plot the time cost vs the nest depth in Figure 2.

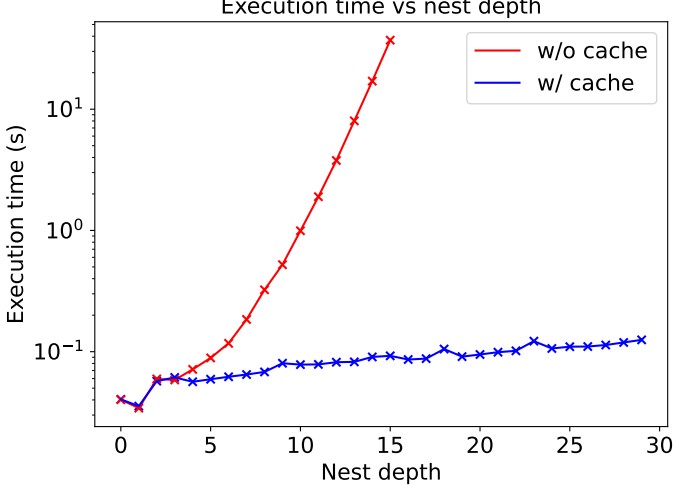

Figure 2: Function call caching significantly improves execution efficiency.

# E  COMPARING GRADIENT ESTIMATORS

Subtracting Equation (19) and (20) we get:

$$\int \left( \frac{\partial I}{\partial \nabla y^\theta(x)} \frac{\partial \nabla y^\theta(x)}{\partial \theta} + \nabla \frac{\partial I}{\partial \nabla y^\theta(x)} \frac{\partial y^\theta(x)}{\partial \theta} \right) dx$$

$$= \int \left( \frac{\partial I}{\partial \nabla y^\theta(x)} \nabla \frac{\partial y^\theta(x)}{\partial \theta} + \nabla \frac{\partial I}{\partial \nabla y^\theta(x)} \frac{\partial y^\theta(x)}{\partial \theta} \right) dx$$

$$= \int \nabla \left( \frac{\partial I}{\partial \nabla y^\theta(x)} \frac{\partial y^\theta(x)}{\partial \theta} \right) dx$$

$$= \nabla \int \left( \frac{\partial I}{\partial \nabla y^\theta(x)} \frac{\partial y^\theta(x)}{\partial \theta} \right) dx = 0 \qquad (26)$$

In Equation (26), the integral evaluate to a constant that is not dependent on $x$; therefore taking $\nabla$ on the integral yields $0$. This proves Equation (19) and (20), when the integrals are discretized, are different estimators of the same quantity.

## F   HIGHER ORDER FUNCTIONAL DERIVATIVE IN LIBXC

```python
def compute(self, inp, output=None, do_exc=True, do_vxc=True, do_fxc=False, do_kxc=False, do_lxc=False):
    """
    Evaluates the functional and its derivatives on a grid.

    Parameters
    ----------
    inp : np.ndarray or dict of np.ndarray
        A input dictionary of NumPy array-like structures that provide the density on a grid and its derivaties. These are labled:
            rho - the density on a grid
            sigma - the contracted density gradients
            lapl - the laplacian of the density
            tau - the kinetic energy density

        Each family of functionals requires different derivatives:
            LDA: rho
            GGA: rho, sigma
            MGGA: rho, sigma, lapl (optional), tau

    output : dict of np.ndarray (optional, None)
        Contains a dictionary of NumPy array-like structures to use as output data. If none are supplied this
        function will build an output space for you. The output dictionary depends on the derivates requested.
        A comprehensive list is provided below for each functional family.
            LDA:
                EXC: zk
                VXC: vrho
                FXC: v2rho2
                KXC: v3rho3
                LXC: v4rho4
            GGA:
                EXC: zk
                VXC: vrho, vsigma
                FXC: v2rho2, v2rhosigma, v2sigma2
                KXC: v3rho3, v3rho2sigma, v3rhosigma2, v3sigma3
                LXC: v4rho4, v4rho3sigma, v4rho2sigma2, v4rhosigma3, v4sigma4
            MGGA:
                EXC: zk
                VXC: vrho, vsigma, vlapl (optional), vtau
                FXC: v2rho2, v2rhosigma, v2rholapl, v2rhotau, v2sigma2,
                     v2sigmalapl, v2sigmatau, v2lapl2, v2lapltau, v2tau2
                KXC: v3rho3, v3rho2sigma, v3rho2lapl, v3rho2tau, v3rhosigma2,
                     v3rhosigmalapl, v3rhosigmatau, v3rholapl2, v3rholapltau,
                     v3rhotau2, v3sigma3, v3sigma2lapl, v3sigma2tau,
                     v3sigmalapl2, v3sigmalapltau, v3sigmatau2, v3lapl3,
                     v3lapl2tau, v3lapltau2, v3tau3
                LXC: v4rho4, v4rho3sigma, v4rho3lapl, v4rho3tau, v4rho2sigma2,
                     v4rho2sigmalapl, v4rho2sigmatau, v4rho2lapl2, v4rho2lapltau,
                     v4rho2tau2, v4rhosigma3, v4rhosigma2lapl, v4rhosigma2tau,
                     v4rhosigmalapl2, v4rhosigmalapltau, v4rhosigmatau2,
                     v4rholapl3, v4rholapl2tau, v4rholapltau2, v4rhotau3,
                     v4sigma4, v4sigma3lapl, v4sigma3tau, v4sigma2lapl2,
                     v4sigma2lapltau, v4sigma2tau2, v4sigmalapl3, v4sigmalapl2tau,
                     v4sigmalapltau2, v4sigmatau3, v4lapl4, v4lapl3tau,
                     v4lapl2tau2, v4lapltau3, v4tau4
```

Figure 3: Many terms are involved in constructing higher order functional derivatives.

## G    EXPERIMENTAL SETTINGS FOR BRACHISTOCHRONE

For the brachistochrone experiment, the initial position is at $(x_0, y_0) = (0, 0)$ and the end position is $(x_T, y_T) = (1, -1)$. The functional we minimize is

$$F(y) = \int_0^1 \sqrt{1 + \nabla y(x)^2} / \sqrt{-y(x)} dx$$

We aim to find

$$y^* = \arg \min_y F(y)$$

We use $y(x) = \text{MLP}(x) \sin(\pi x) - x$ to ensure that it passes through $(0, 0)$ and $(1, -1)$. The MLP is a multi-layer perceptron that maps $\mathbb{R} \to \mathbb{R}$, with hidden dimensions as $\{128, 128, 128, 1\}$. All layers uses the `sigmoid` function as the activation function, except for the last layer which has no activation function.

For the integration, we use a uniformly sampled grid of $50$ points, the starting point is at $0.01$ and the ending point is at $1$. The reason we choose a non-zero starting point is because $0$ is a singular point for the integrand, i.e. the denominator $\sqrt{-y(0)} = 0$. For the optimization process, since it is a toy problem, we use out of the box adam optimizer from optax, with fixed learning rate $1e^{-3}$ and optimize for 10000 steps.

It is worth noting that this is not the optimal setting for doing integration, one can either use a more densely sampled grid or use Monte Carlo integration for a better fitted brachistochrone curve. We choose this setting to show that Equation (19) and Equation (20) are very different estimators, and that Eqation (20) could have a regularizing effect on the scale of functional derivative (Figure 1) due to the fact that functional derivative is explicitly used.

# H  NONLOCAL NEURAL FUNCTIONAL WITH FUNCTIONAL GRADIENT DESCENT

We describe our precedure of optimizing the nonlocal neural functional in more detail here. The neural functional we use has two linear operator layers, the first layer uses a 'tanh' activation function, while the second layer uses no activation. The final output function is compared with the target function via $L_2$ loss. We take a learning rate of $0.1$ for $4$ steps. The code of this experiment is presented below.

```python
import autofd.operators as o

def f(x: Float32[Array, ""]) -> Float32[Array, ""]:
  return jnp.sin(4 * x * jnp.pi)

def b(x: Float32[Array, ""]) -> Float32[Array, ""]:
  return jnp.sin(x * jnp.pi)

def y(x: Float32[Array, ""]) -> Float32[Array, ""]:
  return jnp.cos(x * jnp.pi)

def k(y: Float32[Array, ""], x: Float32[Array, ""]) -> Float32[Array, ""]:
  return jnp.sin(y) + jnp.cos(x)

def layer(k, b, f, activate=True):
  # here k @ f is syntatic sugar for
  # o.integrate(k * broadcast(f), argnums=1)
  g = k @ f + b
  if activate:
    a = o.numpy.tanh(g)
    return a
  else:
    return g

def loss(params, f, t):
  # two layer mlp
  k1, b1, k2, b2 = params
  h1 = layer(k1, b1, f, activation=True)
  h2 = layer(k2, b2, h1, activation=False)
  return o.integrate((h2 - t)**2)

# initialize both k1, k2 to k, b1, b2 to b
param = (k, b, k, b)

# perform gradient steps
l = loss(param, f, t)
print(f"initial loss: {l}")
for i in range(3):
  grad = jax.grad(loss)(param, f, t)
  param = jax.tree_util.tree_map(lambda x, dx: x - 0.1 * dx, param, grad)
  l = loss(param, f, t)
  print(f"loss at step {i}: {l}")
```

As can be seen, in the limited number of steps we take, the loss steadily goes smaller (Figure 4). We visualize the prediction from the neural functional vs the target function in Figure 5, and the kernel $k(x, y)$ from the first layer of the neural functional in Figure 6. The reason we only takes 4 steps of descent is because the learned kernel function gets prohibitively large. We show the JAXPR graphs of $k(x, y)$ from the first neural functional layer for each step in Figure 7, notice that the graph for step 4 is too big that we failed to render it.

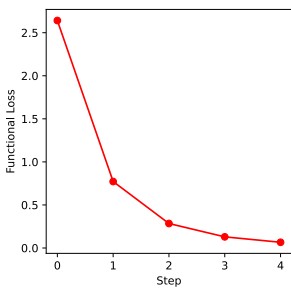

Figure 4: Loss vs Step for the nonlocal functional training

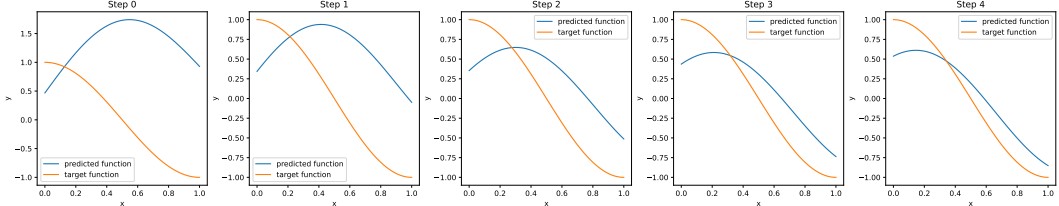

Figure 5: Predicted function vs target function at each step.

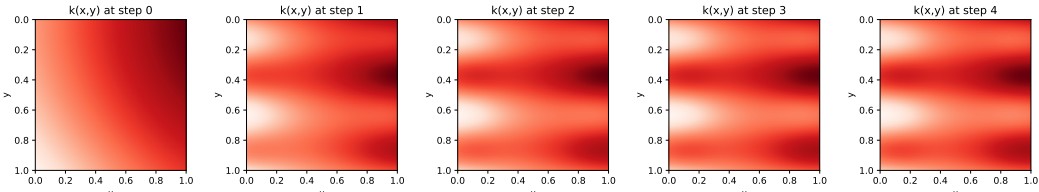

Figure 6: Visualize kernel $k(x, y)$ from the first layer at each step.

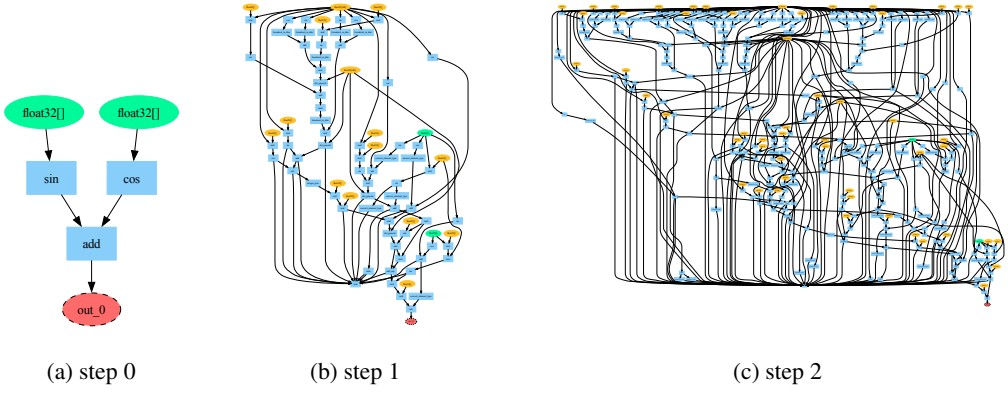

(a) step 0         (b) step 1         (c) step 2

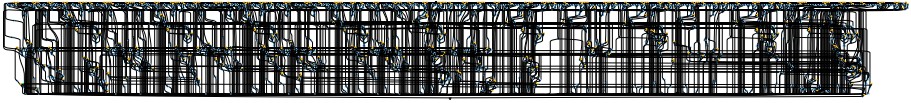

(d) step 3

Figure 7: Jaxpr graph of $k(x, y)$ at each step.

