# OpenReview forum: "Automatic Functional Differentiation in JAX"
_ICLR.cc/2024/Conference — ICLR 2024 poster_

### Official Review · Reviewer_VvN5 · 2023-10-23

**Soundness:** 2 fair
**Presentation:** 2 fair
**Contribution:** 2 fair
**Rating:** 5
**Confidence:** 3

**Summary:**

The authors show that automatic functional differentiation (AutoFD) can be implemented in the same vein as automatic differentiation (AD) in JAX.
The authors introduce operators, namely, compose, ∇, linearize, linear transpose, integrate, JVP, and transpose rules.
The authors provide two applications of AutoFD: Solving the brachistochrone problem and density functional theory.

**Strengths:**

- Automatic functional differentiation is required in many research areas, such as physics, chemistry, mathematics, and machine learning. The topic is very relevant.

**Weaknesses:**

- Presentation is poor (see Questions and Comments below for details) and much more clarification is needed.

- The experiments are not reproducible.

- The baseline of the experiments is too weak.

**Questions:**

# [Question (major)] Infinite dimensional generalization of arrays
In Abstract (and Introduction),
> By representing functions as infinite dimensional generalization of arrays, we seamlessly use JAX’s existing primitive system to implement higher-order functions.
- How was it realized in the proposed method? Could you point the part of the manuscript that concretely explain it?

# [Comment (minor)] Reference
In Introduction,
> To this date, functional differentiation with respect to functions are usually done in one of the following ways: (1) manually derived by human and explicitly implemented; (2) for semi-local functionals, convert it to the canonical Euler-Lagrange form, which can be implemented generally using existing AD tools; (3) use parametric functions and convert the problem to parameter space.
- Could you add some reference papers about these three approaches for readers who are not that familiar with functional differentiation?

# [Comment (minor)] Reference
In Section 3,
> The Schwartz kernel theorem states that any linear operators can be expressed in this form.
- Could you add a reference here? It is more reader-friendly.

# [Question] Complex numbers
In Section 4.2,
> The primitive operators considered in this work are focused to realize the most used types
of operators and functionals described in Section 3.
- [Question] Does your proposed program support complex numbers?

# [Comment (major)] More definitions
In Section 4,
- It would improve the clarity of the present paper to add a rigorous and/or intuitive definition of the Frechet derivative, cotangent space, and transpose rule and also an illustration of the primitive operations used in the present paper, because not all of the readers are familiar with both functional analysis and computer programming. Changing the order of explanations can also be an option (e.g., the begging of Section 4.2.3 can be an intuitive explanation of the Frechet derivative as a generalized directional derivative).

# [Question]
In Section 4.2.1,
> The function inverse on the right hand side of the $T_f (\hat{C})$ rule is not implemented...
- Does it restrict the operations of the proposed AutoFD? Could you take some examples?

# [Question (major)] Grid points
In Section 4.2.5 and Experiment
> We implement the integrate operator by supplying a numerical grid for the integrand function.
- How did you choose the grid points?
- How critical is the numerical error?
- The proposed integral scheme looks like nothing but the conventional numerical integral. Is there any difference?

# [Comment (minor)] Font
In Section 4.3 and Eq. (7--8),
> We denote them as undefined because...
- Changing the font of "undefined" would be good, e.g., mathtt.

# [Comment (major)] Efficienty
- For most of the statements in Section 4.4, I would like to see quantitative results.

# [Questions and Comments (major)] Experiment: Solving variational problem
In Section 5.1, the authors performed an experiment to solve the brachistochrone problem.
- The authors simplify the problem to a parametric fitting of $y^{\theta}(x)$. This is how conventional methods in the numerical analysis of functionals does, as is stated in Section 1 and 2. What does the proposed program enable us to do, or what is the difference from the conventional methods?
- What is the difference between Eq. (17) and (18--19)? Is it whether the Euler-Lagrange equation is used?  If yes, the performance gap given in Figure 2 may come from it.

> It is worth highlighting that the directly minimize in the parameter space is limited by the numerical integration, it easily overfits to the numerical grid.
- Taking random grids in every iteration is often done in learning integrals, which would fill the gap between red and the other curves. I would like to see the performance of such a more rational baseline.

> Better functional optimal are found as can be seen in Figure 2 (right) that the functional gradient is closer to zero.
- The authors should use log scales.

# [Question and Comment (major)] Experiment: Density functional theory
In Figure 3 and 4,
- I could not understand what Figure 4 means, potentially because Figure 4 is not a complete code like Figure 3. Could you provide more details of it?
- Could you clarify how difficult it is to implement higher order derivatives?
- Is Figure 3 simply a wrapper of Figure 4?
- I would like to see numerical results about DFT using AutoFD.

> In the SCF loop of the DFT calculation, ...
- What is SCF?

# [Comment (major)] Detailed experimental settings
- Could you add more details of the experimental settings for reproducibility?

# [Comment (major)] Code submission
- Could you show the code for reproducibility?

# [Question] PDE and FDE
- Can we use the proposed AutoFD to solve a partial differential equations that include functional derivatives?
- Can we use the proposed AutoFD to solve a functional differential equation that include functional derivatives (the task is to get the functional that satisfies the given equation that include functional derivatives)?

# [Comment] Other journals and conferences
- Automatic differentiation is actively discussed in, e.g., MFPS, TOMS, ICFP, TOPLA, POPL, and FoSSaCS. These community might have much more interest in the present paper.


# [Comment (major)] Typos
Please proofread the manuscript before submission.
- In Section 4.1, "...one level of generalization, the term function in..." should be. e.g., "...one level of generalization. That is,  the term function in..."
- In Section 4.2, "... for consistency. i.e. ..." should be "... for consistency; i.e. ..."
- In Section 4.2, ". Which" should be ", which"

---

> ### Author Response · Authors · 2023-11-13
> **Addressing the weakness points and questions**
>
> Thanks for your time reviewing our draft, and the in depth questions on various technical points.
>
> **Q1, infinite dimensional arrays**
>
> This is brought up by more than one reviewers, and we admit that the previous presentation has not give enough background on it. In the updated PDF, we removed all quotes to "infinite dimension", which we find doesn't hurt the presentation. We think it will avoid confusion for the future readers.
>
> Infinite-dimension is not implemented explicitly, we only extend the abstract shape of array to be infinite-dimensional. It is described in 4.1. In the updated PDF, we will stick to generalized arrays. Nevertheless let's explain how infinite dimensional is introduced in the shape.
>
> For a plain array `x` of shape `[5]`, we can retrieve the value at `u` by `x[u]`, where `u` $\in \\{0,1,2,3,4\\}$. We have a finite number of `u`, therefore `x` is finite dimensional.
>
> We extend the shape system to describe shape of a function, e.g. `[f[5]]` is the shape of a function taking an input of a `5` dimensional `float` vector. Mathematically `f[5]` represents a `5` dimensional vector space. Similarly, we can retrieve the value at `u` by calling the function at `u`, i.e. `x(u)`, where `u`$\in R^5$. Since $R^5$ is a continuous space, we have infinite numbers of `u`.
>
> **Q2, could you add some reference papers about these three approaches for readers**
>
> Thank you for pointing it out, we have added references in the updated PDF. We removed the (3) because it doesn't explicitly calculate functional gradient.
>
> **Q3, does your proposed program support complex numbers**
>
> For complex valued function, definitely yes, we are already using it for density functional theory, which deals with wave functions.
>
> For complex functions, it supports as good as JAX does. As many of the operators, `linearize`, `nabla`, `linear_transpose` are already implemented in JAX, we only provide the rules to perform functional differentiation for these operators. So they rely on [JAX's capability](https://jax.readthedocs.io/en/latest/notebooks/autodiff_cookbook.html#complex-numbers-and-differentiation
> ) to support complex functions. `compose` simply does `lambda x: f(g(x))` so it trivially supports complex functions. As for `integrate`, I'm not very familiar with numerical integration for complex functions, but as long as it converts to summation over grids, I suppose there's no difference from real functions.
>
> **Q4, more definition**
>
> We have refactored Section 4.2 to connect the math and programming notations of JVP, VJP and transpose, including the Fréchet notation and our own Fréchet like notation for the linear transpose. We hope it is better this time!
>
> **Q5, does lack of inverse restrict the operations of the proposed AutoFD**
>
> It is technically feasible to support inversion in JAX, we can add inversion rules for each primitive w.r.t each input. However, this seems an independent project because function inversion has much wider use outside of functional differentiation, for example, used in normalizing flows.
>
> One further question would be, what if the function is not invertibe? It then has to be worked out case by case. For example, piecewise invertible functions like `abs` can be separated into invertible pieces to work out the contribution from each piece. For more complex functions, there is no general way of deriving the transpose rule, in my opinion, it is more of a mathematical limitation than a programming one.
>
> **Q6, grid points**
>
> In the integration operator, we provide user the interface for supplying their own numerical integration grids. Our main contribution is on the automatic function differentiation software, there is no numerical/algorithmic contributions here.
>
> There are various APIs for obtaining different grids in existing packages, which can be used with our integrate operator. e.g. [legendre grid in numpy](https://numpy.org/doc/stable/reference/generated/numpy.polynomial.legendre.leggauss.html). However, the users need to case by case decide which grid to use depending on the functions they are integrating, because there is no universal method.
>
> **Q7, font**
>
> We fixed it in the updated PDF.
>
> **Q8, for most of the statements in Section 4.4, I would like to see quantitative results.**
>
> We include a case study in Appendix D where the redanduncy is exponential to the depth of function composition, with function call caching we achieve a linear time cost, while the naive implementation suffers a exponential cost.

---

> ### Author Response · Authors · 2023-11-13
> **Continued**
>
> **Q9, what does the proposed program enable us to do ... is it whether the Euler-Lagrange equation is used?**
>
> Yes the direct approach is not new, in the experiment euler-lagrange approach seems better. What we have contributed here is the capability to use `jax.grad(F)` which automatically gives us the analytical gradient which is the same as derived from euler-lagrange. The advantage compared to explicit doing euler-lagrange are,
> 1. AutoFD handles tensor valued input and outputs automatically.
> 2. With AutoFD we do not need to write our code in the canonical form of euler-lagrange. We can compose functions more flexibly and use `jax.grad` for the functional gradient.
> 3. While euler-lagrange handles semi-local functional introduced in section 3. AutoFD handles non-local functionals via the integrate operator.
>
> **Q10, taking random grids in every iteration is often done in learning integrals**
>
> Taking random points will no doubt reduce the variance of both gradient estimators and make them converge to the same mean. We would like to emphasize that we are not developing a new method or trying to beat any baseline here. The purpose of this example to demonstrate that AutoFD enables us to use a very different gradient estimator via functional derivative.
>
> We've also updated the figure in log scale.
>
> **Q11, confusions on DFT code**
>
> **what Figure 4 (now Figure 5) means:** Figure 4 shows the API of a library called libxc which provides intermediate values for the functional derivative. Traditionally, the functional derivative is achieved by composing the terms listed in Figure 4 using generalized euler-lagrange. Specifically, Vxc, Fxc, Kxc and Lxc corresponds to 1-4th order functional derivatives, right hand side of the colon shows the terms needed for each derivative. What we would like to convey here is that manual construction of functional derivatives of various order is tedious and laborious, while with AutoFD we can simple apply `jax.grad` on functionals.
>
> **Numerical results on DFT:** This is a ongoing work, it is not available for the time being.
>
> **What is SCF:** it is called the self consistent field, we updated it in the draft.
>
> **Q12, could you add more details of the experimental settings for reproducibility?**
>
> Yes we included the experimental settings for brachistochrone in Appendix G. The code is still being reviewed and will be directly open sourced in the near future.
>
> **Q13, PDE and FDE**
>
> Whether this work will benefit PDE and FDE would depend on whether an explicit function gradient is needed. I believe it is case by case. One potential use may be to regularize the functional gradient, like presented in this work: [Neural Integral Functionals](https://openreview.net/forum?id=aQuqw6eVKP)

---

> ### Comment · Reviewer_VvN5 · 2023-11-15
>
> Thank you for the reply, revision, and additional experiment. I also read the other reviews and understand more about the contributions of the paper. They addressed some of my concerns, and the clarity is improved; therefore, I changed Soundness (1 to 2), Presentation (1 to 2), and Rating (3 to 5) accordingly.
> Still, there is a critical problem on clarity.  Let me make some more comments.
>
> [Comment (major)]
> - Some mathematical terms, such as "cotangent" and "Frechet derivative", may still confuse some readers. They have  more rigorous, high-level definitions in math and the terms used in the paper are special cases of them. The abstract, intuitive definitions written in the paper may help readers understand the meaning, but may still keep readers questioning what they are anyway.
>
> [Comment (major)]
> - Please add more details of the experiment; e.g., the number of steps in training, optimizer, learning rate, weight decay, and other hyperparameters, if any.
>
> [Comment]
> - Please add the label of the horizontal axis of Figure 2.
>
> [Comment]
> - Applications in Section 5 are limited to integral functionals. Adding an example of non-integral functionals would be helpful.
>
> [Additional question]
> - Do you have an idea to deal with the functionals whose functional derivative includes the Dirac delta function, which I understand is difficult in general?
>
> [Typo]
> - In Appendix E, "... that is not dependent on x, therefore ..." should be "... that is not dependent on x; therefore ..."

---

> > ### Author Response · Authors · 2023-11-17
> > **Adding AutoFD for nonlocal functional**
> >
> > Thank you for the feedback and further questions, we address them as follows
> >
> > **rigorous, high-level definitions**
> >
> > Under the context of this paper, I would choose not to go further in the mathematical definition. Part of the reason is that I'm not an expert in differential geometry of functional analysis, I may not connect them correctly in the automatic differentiation (AD) context. I would also argue that introducing them here is too heavy, in the context of implementing AD, it is more of a naming, may be even the difference between tangent and cotangent is not that important according to Section 1 of https://arxiv.org/abs/2105.09469. Let us know if you disagree on this.
> >
> > **more details of hyperparameters**
> >
> > Our apologizes for overlooking these numbers, they are added now.
> >
> > **non-integral functionals**
> >
> > Do you actually mean nonlocal functionals which Euler-Lagrange can't handle? We introduced one more experiment section that tests our capability of computing functional gradients for nonlocal functionals. We implemented a high-level function generalization of MLP, which takes a function as an input, and outputs another function. The trainable part of such functional are the kernel and bias functions in the linear layer. We use AutoFD to perform gradient descent directly in the function space to demonstrate our capability in dealing with nonlocal functionals. The additional results can be found in Section 5.3 and Appendix H.
> >
> > **Dirac delta function**
> >
> > Currently we can't do it, since our approach is based on the same idea of automatic differentiation, rather than depending on symbolic math.
> >
> > **Minor issues and typos**
> >
> > We will fix them in the next revision.

---

> ### Comment · Reviewer_VvN5 · 2023-11-21
>
> Thank you for the changes to the paper.
>
> **rigorous, high-level definitions**
>
> I understood your situation. Then, I think adding reference papers/texts to immediately after the math terms is a simple way to go. How about it?
>
> **non-integral functionals**
>
> I literally meant non-integral functionals, e.g., $F([f]) = \frac{\int dt \sqrt{1 + {f^\prime}^2(t)} }{\int dt \sqrt{1 + f(t)} }$ (even a toy problem is OK), but anyway, the extra experiment in Appendix H is also a nice, practical example, and it made the paper more convincing and helpful for users.

---

> > ### Author Response · Authors · 2023-11-21
> >
> > **reference to the formal definitions**
> >
> > Good point, we updated accordingly in Section 3.2.
> >
> > **Non-integral functionals**
> >
> > Ah, thanks for clarifying. I was thinking of things like $f\mapsto max_x f(x)$ which is nontrivial under the set of operators we introduced. For now we don't have a realistic example in mind yet that gives a non-integral functional like the above. In general, we can differentiate as long as every building block is differentiable. For example, the above functional is made of two integral functionals and a `jnp.divide` operation. The former is available in AutoFD and the latter is in plain JAX. We'll include this example as a unittest when we release the code. Also, thanks for the appreciation of Appendix H.

---

> > > ### Comment · Reviewer_VvN5 · 2023-12-02
> > > **To Authors, Reviewers, and Chairs**
> > >
> > > Thank you for your time and effort. The authors' thorough revision much improved the paper quality.
> > > In view of it, although my Rate is 5 (marginally below the acceptance threshold), my stance is almost neutral, and I find it not surprising at all if this paper were to be accepted.

---

### Official Review · Reviewer_RHkE · 2023-11-01

**Soundness:** 3 good
**Presentation:** 2 fair
**Contribution:** 3 good
**Rating:** 6
**Confidence:** 2

**Summary:**

In this paper, the authors extend the JAX automatic differentiation system to support higher-order derivatives -- i.e. derivatives over functions (often called functionals), rather than arrays.

Some background: the core JAX autodiff system is based on a first order functional programming language.  The only data type that it supports is the Array (or tuples/dictionaries of Arrays), and all of the built-in differentiable primitive operations are first-order functions over arrays.  Consequently, JAX autodiff can only compute gradients with respect to Arrays.

It is important to note that the larger JAX library has plenty of higher-order API calls (e.g. vmap, linearize, jvp etc.), and python itself supports higher-order functions.  However, the core autodiff system works by *tracing* python functions to a first-order computation graph, and then computing gradients on that graph.

In this paper, the authors extend JAX type system to support functions as first-class citizens in the computation graph, and they introduce a set of differentiable primitive operations that are higher-order (e.g. function composition).  The extended autodiff
system uses the same JAX autodiff machinery, but it can compute gradients with respect to functions/functionals as well as Arrays.

The authors describe the differentiation rules for a core set of higher-order primitive operations, and give a few examples, primarily drawn from engineering and physics, for doing autodiff over functionals.

**Strengths:**

The paper is reasonably well-written, and the idea is mathematically interesting.

**Weaknesses:**

Although the paper is reasonably well-written, I had a very hard time following it.  Part of the problem is my fault.  Although I have deep knowledge of both JAX and automatic differentiation systems in general (having implemented several of them myself), I am not
particularly familiar with the underlying mathematics of functional analysis, Fréchet derivatives, and so on, which are used in this paper.   I have not checked the math, and I am happy to defer to other reviewers who have a deeper understanding.

Since this paper is being submitted to ICLR, rather than a computational mathematics conference, I had expected the authors to provide a gentle introduction to some of the underlying concepts, suitable for ML practitioners.  Sadly, they do not, and I suspect that most ICLR readers will have the same problems understanding it that I did.

The authors claim that "functions are represented as infinite dimensional generalizations of arrays", but they do not explain how, nor do they even cite any source for this claim.  (Perhaps a functional analysis textbook?)  Moreover, I assume the "infinite dimensional array" is simply a mathematical abstraction, and it is unclear to me why it is even relevant.  In the body of the paper, functions actually seem to be represented in the usual way as symbolic programs.  As one would expect from a practical algorithm, infinite dimensions do not appear.

The actual differentiation rules are written in terms of forward derivatives and *transpose* operations.  The use of transpose is due to recent work by Radul et al., but will likely be unfamiliar to ML practitioners who are used to traditional backpropagation.  It would have been helpful if the authors had given an example of how these two operators work in a conventional (non-higher-order) setting, before diving into the higher-order case.

Even after doing my best to read this paper carefully, I am still unsure about how higher-order functions (functionals) are actually defined and represented in the core language. The differentiation rules given here do not seem to constitute what I would consider to
be a core programming language.  E.g. the authors provide differentiation rules for function composition, but not for function application or function definition.  That leads to me believe that the code for functionals would have to be defined in a point-free functional programming style, as is used by some other autodiff systems in the literature.  However, the examples in the paper just show ordinary python.  Is the python code traced, and then translated into the primitive operations, as is usual for Jax?  What are the details of this translation, since going from python to a point-free representation is not necessarily trivial?

The differentiation rules for function composition $f \circ g$ require that $g$ is invertible.  If composition is the basic mechanism used to build complex programs (as is usually the case in point-free languages), then this would seem to be a very severe limitation.

My final point of confusion is that in most cases of practical interest, the functional that we are computing a derivative for is really just an ordinary symbolic function that is parameterized by some array $A$.  The value that we actually want to solve for is $A$ -- the higher-order derivative is just an intermediary step.  In this situation, ordinary Jax works just fine without higher-order automatic differentiation; the higher-order operations are eliminated by tracing and partial evaluation, and the first-order autodiff system then solves for the gradient of $A$.

The authors actually allude to this fact, but do not provide a detailed discussion of the tradeoffs between solving directly for $A$ using tracing and first-order autodiff, and doing something more complicated with higher-order autodiff.  Given the various limitations and restrictions on the higher-order methods, the former seems decidedly simpler.  Why would I want to use this system?  Can you explain why some problems can't be solved with ordinary Jax?

Finally there are some typos which added to my confusion.  E.g. page 5 second paragraph $C(x, y \to x + y, f, g)$ needs extra parens: $C((x, y) \to x + y, f, g)$ otherwise it makes no sense.

**Questions:**

Please see weaknesses, above.

---

> ### Author Response · Authors · 2023-11-13
> **Addressing the weakness points and questions**
>
> Thanks for your time reviewing our draft and the in depth comments on various technical points.
>
> **Weakness 0, hard to follow mathematics of functional analysis, Fréchet derivatives**
>
> We refactored Section 4.2 to connect the math and programming notations of JVP, VJP and transpose, including the Fréchet notation and our own Fréchet like notation for the linear transpose. We hope it is better this time!
>
> **Weakness 1, Confusion on infinite dimensional array**
>
> This is brought up by more than one reviewers, and we admit that the previous presentation has not give enough background on it. You're right that the infinite-dimensional array is only abstract. In the updated PDF, we removed all quotes to "infinite dimension", which we find doesn't hurt the presentation. We think it will avoid confusion for the future readers.
>
> Nevertheless, let's also explain why the generalized array described in 4.1 is infinite-dimensional.
> For a plain array `x` of shape `[5]`, we can retrieve the value at `u` by `x[u]`, where `u` $\in \\{0,1,2,3,4\\}$. We have a finite number of `u`, therefore `x` is finite dimensional.
>
> We extend the shape system to describe shape of a function, e.g. `[f[5]]` is the shape of a function taking an input of a `5` dimensional `float` vector. Mathematically `f[5]` represents a `5` dimensional vector space. Similarly, we can retrieve the value at `u` by calling the function at `u`, i.e. `x(u)`, where `u`$\in R^5$. Since $R^5$ is a continuous space, we have infinite numbers of `u`.
>
> **Weakness 2, introduce Radul et al**
>
> Thanks a lot for this suggestion, In our refactored Section 4.2 we show how JVP can be converted to VJP via a linear transpose.
>
> **Weakness 3, how the math map to core programming language**
>
> We provide a minimal but self contained code snippet for the nabla operator on scalar functions in Appendix A, in the hope to provide a better overall understanding of the system.
>
> As for the comments on the point free style, I'm not sure I fully understand, I guess
>
> ```python
> def nabla(f):
>   return jax.grad(f)
> ```
>
> is somehow point free because we don't pass any argument to `f`? However, when we apply `nabla` on a function, we pass that function to `nabla`, thus it is not point free. Therefore, the operators are implemented in the non-point free plain python code, but the functions passed into the operator are used in a point free style.
>
> **how a function is defined**: We registered a custom function to abstract value converter at `jax.core.pytype_aval_mappings[types.FunctionType]`, which enables JAX to convert a function into a abstract array that we defined during tracing. This is added to section 4.1
>
> **Weakness 4, invertibility seems a severe limitation**
>
> It is technically feasible to support inversion in JAX, we can add inversion rules for each primitive w.r.t each input. To invert a function, we first trace the function into JAXPR and transform the JAXPR graph by applying the inverting rule on each primitive. However, this seems an independent project because function inversion has much wider use outside of functional differentiation, for example, used in normalizing flows.
>
> One further question would be, what if the function is not invertibe? It then has to be worked out case by case. For example, piecewise invertible functions like `abs` can be separated into invertible pieces to work out the contribution from each piece. For more complex functions, there is no general way of deriving the transpose rule, in my opinion, it is more of a mathematical limitation than a programming one.
>
> **Weakness 5, are there problems not solvable with ordinary JAX**
>
> Yes, this work is originally motivated by density functional theory (DFT) in quantum chemistry. In DFT, it often happens that we need the functional derivative explicitly, for example, the loss function could be in the form of $L(\theta) = \int \frac{\delta F}{\delta f}(x) g^\theta(x)dx$. Given the functional $F$, it is not very trivial how to write this loss directly in JAX. It is usually hand derived and implemented, like how bp is done without automatic differentiation in the old days. By AutoFD, we would like to offer the "What You See is What You Get" (WYSIWYG) style of programming for problems where functions are first class citizens, e.g. quantum physics deals with wave functions.
>
> **Weakness 6, Tradeoffs between two differentiation methods**
>
> We bring up the different methods to demonstrate that functional derivatives could provide a different angle when we solve the same problem. In the brachistochrone problem it happens to be better via functional derivative, hypothetically, it brings us closer to the minimal in the space of function than in the space of parameters. However, further theoretical investigation is needed to reach a conclusion whether it is general to all problems. My feeling is that most likely it is case by case depending on the variance of the estimator, therefore we didn’t expand further on this topic.

---

> > ### Comment · Reviewer_RHkE · 2023-11-17
> >
> > Thank you for the changes to the paper -- the additions to the introduction do indeed improve the readability.
> >
> > In answer to your question about the "point free" programming style, it is covered in Elliott's "The simple essence of automatic differentiation" https://arxiv.org/abs/1804.00746, which you cite.  (Note that I am not affiliated with Elliott or his work.)  See also Hughes' "Generalizing Monads to Arrows" and "Programming with Arrows" for a broader introduction to the history of "point free" programming.
> >
> > To summarize -- a typical formal programming language (e.g. the lambda calculus) has special syntax for introducing functions (e.g. f = lambda x: ...) and for applying functions (e.g. f(x)).  Moreover, the body of a function can refer to intermediate values via named variables (e.g. x).  This syntax is difficult to work with directly, so AD systems like Jax trace these functions to construct a computation graph.  Tracing strips out the variable names -- named intermediate values simply become edges in the graph.  The actual AD algorithm (forward or backward) is then usually expressed as an algorithm over graphs.
> >
> > "Point-free" programming is an alternative syntax for specifying programs, in which larger functions are constructed by composing smaller functions together, using a core set of primitive operators.  There is no syntax for lambda, nor are there named variables (the variables are "points", hence "point-free").  Sequential composition (e.g. $f \circ g$) is the most familiar operator, but there are others, some of which you mention in your paper -- parallel composition (which you call zip), and other operators for rearranging arguments.  These primitive operators are drawn from category theory.  In essence, the set of primitive composition operators are an algebra for specifying computation graphs.
> >
> > The advantage of using an algebra, instead of a computation graph, is that the algebra exposes certain symmetries, and in so doing it vastly simplifies the AD algorithm.  In the category-theoretic formulation, forward and backward AD can be seen as duals of each other, while this symmetry is not obvious in the graph-based presentation.  Applying this technique, however, usually involves a step akin to tracing, in which the normal lambda-calculus based language (which is easy for humans to read) is transformed into the point-free form (which is easy to do AD in).  Elliott performs this transformation in Haskell via a compiler extension; but you don't mention how you do it in python.
> >
> > Your work is strongly reminiscent of the category-theoretic treatment of AD, especially since you include many of the same operators.  However, although you cite Elliott's work, you don't draw the connections to it that I just did.  The "completeness of the primitives" section seems to be missing a few important primitives, e.g. duplication.  Moreover, you also don't describe any automatic translation from python functions to the category-theoretic form.  Do you have to apply operators like zip by hand, and then rely on JAX tracing to translate from python?  That would result in a weird half-way representation, where you have both a computation graph, and an algebraic structure embedded in it, with AD operating on both...
> >
> > In addition, I admit that I am still confused as to why function inversion is necessary in your formulation; no such inversion is necessary in other formulations of AD that I have seen (e.g. compare your rule for function composition to Elliott's).  Since functions are not automatically invertible, this would seem to be a major sticking point, and I think it warrants further discussion.

---

> ### Author Response · Authors · 2023-11-18
> **Relation to Elliott's work**
>
> Thanks for the further comments. I think we're aligned on the form of "point free" programming. On the comment about the connections to Elliott's work, we indeed drew a connection in the related work section,
>
> > "There are a number of works that studies AD in the context of higher-order functionals (Pearlmutter & Siskind, 2008; Elliott, 2018; Shaikhha et al., 2019; Wang et al., 2019; Huot et al., 2020; Sherman et al., 2021). They mainly focus on how AD can be implemented more efficiently and in a compositional manner when the code contains high-order functional transformations, which is related to but distinct from our work."
>
> In short, we believe they are related but distinct topics, let me explain why using the composition operator.
>
> As you mentioned we can describe $f$ in lambda calculus `f = lambda x: ...`, where `x` **is a point in the domain of** `f`. In the same vein, we can describe the $\circ$ as $\circ=\text{lambda}\\;f,g: f\circ g$, where $f$ **is a point in the domain of** $\circ$. In the paper, we call $\circ$ an operator, or higher order function. Here let's be more specific and call $f$ a **first order function** and $\circ$ a **second order function**, where the inputs to the second order function are first order functions. I believe in the context of your discussion, point free is relative to the first order function, because $x$ never appears, but it is not point free to the second order $\circ$ because $f$ and $g$ are its points.
>
> Now let's expand on the distinction from Elliott's work, let's use $D(f)(x)$ to denote differentiating $f$ at the point $x$. In short, Elliott's studies $D(f\circ g)(x)$ while ours studies $D(\circ)(f)$.
>
> - **Elliott's work** studies how to **perform AD for first order function** more efficiently when your program has second order functions. Concretely in the example of composition, how to calculate $D(f \circ g)(x)$. As you mentioned, although we can naively trace $f\circ g$, it would be more efficient if we know the algebraic relation $D(f\circ g)(x) = D(f)(g(x))\circ D(g)(x)$ (Theorem 1 in Elliott's).
>
> - **Our work** studies how to **perform AD for second order function** in the sense of variational calculus. Again in the composition example, we study how $D(\circ)(f)$ and $D(\circ)(g)$ can be computed automatically. Let's call it **Automatic Functional Differentiation (AFD)** instead of AD.
>
> To this point, I hope it is clear how AFD can be implemented using the same mechanism in JAX, as one can find the resemblance between $D(\circ)(f)$ and $D(f)(x)$. More specifically, in python AFD does the following
>
> ```python
> def F(f, g):
>   return integrate(compose(f, g))
> dFdf = jax.grad(F, argnums=0)(f, g)
> ```
> Although the line `integrate(compose(f, g))` is point free in your definition because there is no `x` fed to `f` or `g`, it is not point free for the second order function `F`. Appendix A provides more example on how we implement an operator in python, again the code is point free in first order, but not point free in second order.

---

> ### Author Response · Authors · 2023-11-18
> **Completeness of primitives & Inversion**
>
> **Completeness of primitives**
>
> We explained what we meant by completeness in 4.3
> > We can inspect whether these rules forms a closed set by examining whether the right hand side of the rules (7) to (18) can be implemented with these primitives themselves.
>
> Namely, because the JVP and transpose rules of operator A could depend operator B, we checked and implemented a closed set such that any operator in the set only depend on the other operators in this set. We didn't mean that the set of operators we introduced in the paper is complete to implement any functionals. For example, the functional $f\mapsto\max_x f(x)$ would be nontrivial to implement and differentiate, and can't be expressed with the operator primitives we introduced.
>
> I tried to find duplication from Elliott's work, it seems to me that $dup: \lambda a\rightarrow(a,a)$ is a first order function instead of an operator (second order function).
>
> **Inversion of function**
>
> Inversion of a function is not necessary for AD (differentiating first order functions), however when we generalize AD to AFD (differentiating second order functions), the summations is generalized to integration. For example, the matrix vector product $Mv$ is generalized to $\int M(x,y)v(y) dy$.
>
> In first order functions, the transposition of a linear function $x\mapsto Mx$ is $y\mapsto M^\top y$, we can see the relation that $\langle Mx, y\rangle=\langle x, M^\top y\rangle=\sum_i (Mx)_iy_i$. When generalized to operators, we use $\hat{O}^*$ to denote the transposition (adjoint) of an operator $\hat{O}$, it also needs to satisfy $\langle \hat{O}f, g\rangle=\langle f, \hat{O}^*g\rangle=\int (\hat{O}f)(x)g(x) dx$. Notice that the former involves a $\sum$ which is generalized to $\int$ in the latter. The function inverse comes out when we change the coordinate of integration, see Appendix C1 for more details.

---

> ### Comment · Reviewer_RHkE · 2023-11-20
>
> These are both excellent explanations, thank you.  Is there any way you can work them into the text of the paper?
>
> In particular, other readers who are familiar with Elliott's work (as I am) might also become confused when they see AD over what /seem/ to be a very similar set of operators being defined in very different ways.
>
> With respect to dup, it is indeed a first-order operator, but it's important because (1) it is not usually treated as an operator in graph-style AD, but (2) its derivative becomes (+) in reverse-mode, which is one of the main sources of complexity and symmetry-breaking in the graph-style AD algorithms, so its inclusion is absolutely required.
>
> I am wondering if something similar might not also be true in your setup.  It might also help with the common-subexpression-elimination problem that you discuss; AFAIK, the example you give could be easily resolved by using dup in an appropriate way.

---

> > ### Author Response · Authors · 2023-11-21
> >
> > **Add above discussion in the paper**
> >
> > We're glad that it helps, thank you for the questions that triggered this discussion. It is indeed necessary to include the explanation on the relation to Elliott's to provide a better context. We shift the related works to the latter part of the paper so that we can include the above discussion with math symbols.
> >
> > **CSE with dup**
> >
> > I think I get your point here, basically $c_1 = f\circ h + g\circ h$ is equivalent to $c_2 = \text{zip}(f,g)\circ \text{dup}(h)$; while invoking $c_1$ triggers $h$ twice, $c_2$ removes this redundancy. However, we do still want the simplicity of syntax in the former, therefore we need an optimization stage that performs graph rewriting to convert the former to the latter.
> >
> > As we perform AFD on second order functions, the first step of which is to trace the JAXPR at the second order. Having access to this second order JAXPR graph brings an interesting and useful side effect, that we can introduce a graph rewriting phase that performs various kind of algebraic optimizations. For example, replacing $f\circ h + g\circ h$ with $\text{zip}(f,g)\circ \text{dup}(h)$, or replacing $D(f\circ g)(x)$ with $D(f)(g(x))\circ D(g)(x)$. We agree with you that systematically supporting JAXPR rewriting is more general than the function call caching approach used in this work.
> >
> > With the above said, we like to emphasize that the **main contribution** of this work is still on how to **differentiate** the second order JAXPR graph as opposed to how to **perform algebraic optimization** on the this graph. The latter is definitely a worthy topic itself that may lead to a future work.
> >
> > **Implementation of dup**
> >
> > We can alias it with
> > ```python
> > def dup(f):
> >   return compose(lambda x: (x, x), f)
> > ```
> > Would be happy to include it in our open source code.

---

### Official Review · Reviewer_4ztS · 2023-11-02

**Soundness:** 3 good
**Presentation:** 3 good
**Contribution:** 2 fair
**Rating:** 8
**Confidence:** 4

**Summary:**

This paper introduces functional differentiation (i.e., derivatives of functionals, which are functions that take other functions rather than values as inputs). The package directly builds on JAX's automatic differentiation machinery, introducing a new datatype for functions (section 4.1) and then implementing the appropriate linearization and transposition rules for each primitive operation. The primitive operations implemented are differentiation (nabla), linearization (Fréchet derivative), linear transposition, and integration, as well as some utility operators (composition, permutation, zipping). Caching is used to perform common subexpression elimination while the computation graph is being built. Finally, some experiments using functional differentiation are performed (brachistochrone problem, exchange-correlation functionals) as showcase.

---

Updated my rating from 6 to 8 in response to the changes during the discussion period.

**Strengths:**

I think this paper is quite straightforward, but not in a bad way: The text is clearly structured, and a good balance is found between the theory behind and the implementation of the framework. I appreciate how the authors were able to re-use JAX's machinery, which allows them to benefit from a lot of JAX's strength (compilation backends, debugging tools, etc.) and the simplicity of the brachistochrone in code is quite compelling.

**Weaknesses:**

The main weaknesses of this paper are some of the limitations discussed in section 6. In particular, not having any approach for function inversion seems like a shortcoming.

**Questions:**

Could the machinery used to register derivatives not also be used to register function inverses?

I would also love to see a more thorough discussion of the integration trade-offs: How should the user know what numerical grid to provide? And how sensitive should they expect the outcome to be to the grid provided?

As this is a software package, I would also like to know some of the details regarding public release: Will the code be on GitHub (or some other platform)? Under what license? Will there be documentation, tutorials, or notebooks? Does the code take the form of a separate Python package, or is it a fork of JAX?

---

> ### Author Response · Authors · 2023-11-13
> **Addressing the weakness points and questions**
>
> Thank you for your time spent reviewing this draft, the positive acknowledgement of our contribution, and the suggestions.
> We address each point as below.
>
> **Weakness 1 & Q1, function inversion not implemented**
>
> It is technically feasible to support inversion in JAX, we can add inversion rules for each primitive w.r.t each input. To invert a function, we first trace the function into JAXPR and transform the JAXPR graph by applying the inverting rule on each primitive. This seems an independent project because function inversion has much wider use outside of functional differentiation, for example, used in normalizing flows.
>
> **Q2, how to supply numerical grid**
>
> To our knowledge, the Fourier, Legendre grids mentioned in Section 4.2.5 are two of the most used numerical grids. For example, they are used in the recent [state space model for time series](https://arxiv.org/abs/2008.07669). They are exact when used with linear sums of planewaves / polynomials respectively. However, they are also used for other functions, which is equivalent to expanding the functions in planewaves/polynomials with a truncation error.
>
> **Q3, release of code**
>
> The source code will be open sourced under Apache license, it will be provided as a standalone python package. Documentation and tutorials will be provided too.

---

### Official Review · Reviewer_Mbhg · 2023-11-04

**Soundness:** 4 excellent
**Presentation:** 2 fair
**Contribution:** 3 good
**Rating:** 8
**Confidence:** 4

**Summary:**

In this work the authors introduce a package for the machine learning framework JAX, to enhance JAX with functional differentiation, stemming from functional calculus / the calculus of variations. This enables the easier expression of e.g. neural operators, as they implicitly rely on the functional framework. To this end, the authors introduce 5 new language primitives, namely `compose`, `nabla`, `linearize`, `linear transpose`, and `integrate`. All implemented in pure Python, and complemented by the theoretical derivation of the individual operators.

The introduced extension is subsequently validated on examples from particle dynamics with the brachistochrone problem, and the exchange-correlation functional stemming from density functional theory.

**Strengths:**

Where this paper shines is its connection with a very strong theoretical background stemming from functional calculus, and how it derives its proposed extensions to JAX from said theoretical motivation. The two chosen examples only underline this further.

It removes previous constraints from a machine learning framework, hence acting in large part as an enable of future work using functional calculus, such as for learned operators.

**Weaknesses:**

At the same time, the work suffers from a number of unclear treatments of JAX, and a failure to establish the usage of the proposed framework for neural operators, such as the Fourier Neural Operator.

JAX as a framework:
- The authors explain the tracing into a DAG, and the mapping of primitives to XLA, the compilation backend underpinning JAX. What they miss in this instance though is the intermediate layer of JAXPR, JAX's internal representation, and the mentioning of operation-tracing, and XLA-compilation, only leads to a lack of clarity. I'd suggest to add a diagram of the pieces of JAX's architecture you rely on, and remove mentions of XLA-mapping of ops etc. from the latter parts of the paper.
- While provided for some introduced operations, a number of operations do not have their implementation code attached to them. The addition of the code in the main paper, or the appendix would contribute greatly to further the clarity of the exposition.

Neural Operators:
- While neural operators, and most specifically Fourier Neural Operators, are presented as a clear motivation for the work, they are sadly not used in the experimental evaluation of the presented extension. Addition of a functioning Fourier neural operator based on `autofd` would significantly strengthen the paper's claims. Evaluation of a Fourier neural operator could for example be in the form of a normal JAX-based implementation, and an `autofd`-using implementation, where the code could for example be much more succinct with `autofd` while matching the performance of the JAX-native implementation.

In addition the draft has a number of minor typos, the addressing of which would improve the legibility greatly. For example:
- Page 1, last paragraph: "Base" -> Based
- Page 2, last line: "maps arbitrary" -> map arbitrarily

**Questions:**

A number of questions arose while reading through the paper:

- Did you perform an analysis of the computational properties of AutoFD, and most specifically the way JAX compiles AutoFD code if it is not being mapped to XLA-ops?
- Is there an implicit trade-off computationally or conceptually to the chosen representation as an infinite-dimensional array?
- Why did you choose a purely Python-based implementation of AutoFD, as compared to a version of AutoFD acting directly on the JAXPR?
- For the _Efficiency of Implementation_, have you considered tracing your implementation with perfetto
- How do you anticipate an inversion operator in JAX to be feasible? Would it be possible to implement such operator, if you were writing the operators at the level of the JAXPR?

---

> ### Author Response · Authors · 2023-11-13
> **Addressing the weakness points and questions**
>
> Thank you for your time spent reviewing this draft, the positive acknowledgement of our contribution, and the suggestions. We address the individually as follows
>
> **Weakness 1, introduce JAXPR instead of XLA:**
>
> We bring up XLA mainly in section 4.4 to mention that optimizations like common subexpression elimination (CSE) is only performed at the XLA level, and the python level lacks this mechanism. We think it is necessary for motivating function call caching. Although the primitives we implement has strong ties to JAXPR, the actual implementation (We added in Appendix A) does not require the knowledge of JAXPR. We think it would be too heavy to introduce to this level.
>
> For the overall connection with JAX, we refined Section 4.2 our updated PDF, where we first introduced the mechanism of JAX and how we extend to higher order functions.
>
> **Weakness 2, provide code implementation**
>
> We’ll make the code open source in the near future. We also include a minimal but self contained example in Appendix A in our updated PDF to provide better clarity on how the math are mapped to code.
>
> **Weakness 3, FNO implementation with AutoFD**
>
> Thanks for bringing this up, along the way we actually thought of whether FNO is a good demonstration. It is true that we can implement FNO with a more succinct syntax, e.g. directly algebraic operations on functions. However, this syntactic simplicity does not relate directly to functional derivative. In the case of FNO, once we construct the forward computation, we then differentiate only with respect to the parameters, which deviates from our major contribution of differentiation with respect to functions. Therefore we focus on examples where functional derivatives makes a difference.
>
>
> **Q1, analysis of computational properties:**
>
> Yes, since our primitive operators are in python, they lack the equivalence of common subexpression elimination (CSE) in XLA, we therefore introduced function call caching for the remedy. It is discussed in section 4.4. In our updated PDF, we include a case study in Appendix D where the redanduncy is exponential to the depth of function composition, with function call caching we achieve a linear time cost, while the naive implementation suffers a exponential cost.
>
> **Q2, infinite-dimensional array:**
>
> The infinite-dimensional array is only abstract, based on the reviews, we decided to remove the quotes to "infinite dimension" in our rebuttal pdf, which doesn't hurt the presentation. We think it will avoid confusion for the future readers.
>
> Nevertheless, let's also explain why the generalized array described in 4.1 is infinite-dimensional.
> For a plain array `x` of shape `[5]`, we can retrieve the value at `u` by `x[u]`, where `u` $\in \\{0,1,2,3,4\\}$. We have a finite number of `u`, therefore `x` is finite dimensional.
>
> We extend the shape system to describe shape of a function, e.g. `[f[5]]` is the shape of a function taking an input of a `5` dimensional `float` vector. Mathematically `f[5]` represents a `5` dimensional vector space. Similarly, we can retrieve the value at `u` by calling the function at `u`, i.e. `x(u)`, where `u`$\in R^5$. Since $R^5$ is a continuous space, we have infinite numbers of `u`.
>
> **Q3, why pure python instead of JAXPR**
>
> We need to clarify that we do capture higher order functions as JAXPR, but unlike ordinary JAXPR whose primitives consumes arrays (and thus can be lowered and compiled), in our JAXPR the primitives are operators that consumes python functions, therefore the operators themselves have to be python functions. It would be easier to understand by the minimal code example in Appendix A.
>
> **Q4, tracing with perfetto**
>
> The profiling tools and visualization tools like perfetto is more suitable for analysing the compiled program. As mentioned above, we have a case study that measures the efficiency in Appendix D.
>
> **Q5, inversion operator in JAX**
>
> It is technically feasible to support inversion in JAX, we can add inversion rules for each primitive w.r.t each input. We first trace the function into JAXPR and transform the JAXPR graph by applying the inverting rule on each primitive. This seems an independent project because function inversion has much wider use outside of functional differentiation, for example, used in normalizing flows.

---

> ### Author Response · Authors · 2023-11-16
> **Update: we added neural operator experiment**
>
> In the new revision PDF, we updated our experiment section with an additional section that tests AutoFD on an MLP-like neural operator. For simplicity we choose a uniform grid for integration but does not exactly follow the details of FNO. We give the operator a target function to fit, and use the $L_2$ distance between the prediction and the target as the functional loss.
>
> As discussed in the weakness 3, usually neural operator is trained by gradient descent directly in parameter space.
> In our experiment, we perform gradient descent in the function space, by subtracting the kernel and bias functions with their functional gradient. For example, we start with a very simple kernel function $k(x,y)$, then we perform $k(x,y)\leftarrow k(x,y)-\eta\frac{\delta L}{\delta k}(x,y)$, now $k(x,y)$ gets more complex, causing the next $\frac{\delta L}{\delta k}(x,y)$ to be even more complex. In Figure 9 we show how the JAXPR graph of $k(x,y)$ grow over time. Due to this compounding effect, we can only perform 4 step of gradient descent. However, in these 4 steps, we can see the loss are decreasing, and the prediction gets closer to the target function.
>
> This new experiment is mainly design to show that AutoFD handles nonlocal functionals very well, and can correctly deal with procedures like above that leads to highly complex functions. By no means do we suggest training neural operator in this way, as discussed in weakness 3, FNO may not be the most suitable problem for AutoFD.

---

> > ### Comment · Reviewer_Mbhg · 2023-11-23
> > **Major weaknesses addressed, as well as many further improvements**
> >
> > Reading through all the responses, and reading the revision I thank the authors for all their effort, and will raise my score accordingly.

---

### Author Response · Authors · 2023-11-13
**Reply to all reviewers**

We thanks all the reviewers for their precious time, the positive acknowledgements of our work and the feedbacks that will help us improve.

In the rebuttal revision PDF, we have made the following changes (highlighted in red in the PDF),

1. We removed quotes to "infinite dimensional" as it turns out to be confusing to the readers. We find that removing it doesn't hurt the understanding of the paper.
2. We refactored section 4.2 to give a more friendly introduction to readers who's not familiar with
    - how JVP/transpose rules are used in JAX for computing gradients.
    - the meaning of the Fréchet derivative and the similar transpose symbol that we introduced.
    - what are the variables that we call primal/tangent and cotangent.
3. We add a minimal but self contained code showing how a primitive operator is implemented, and how their JVP/transpose rules are defined and used.
4. We update 4.4 with a supporting case study. The results are added in Appendix D, showing function caching effectively reduces the redundancy.
5. We also give a more detailed experimental setting in Appendix G for reproducibility.
6. Finally, we fixed the typos and added references where necessary.

We address each reviewer's questions by replying separately.

---

> ### Author Response · Authors · 2023-11-16
> **Adding experiments for nonlocal functional**
>
> In the new revision, we include experiment on neural operator which is a generalization of MLP to higher order. We have trained it using gradient descent directly in the function space, and show results in Appendix H.

---

### Author Response · Authors · 2023-11-20

Dear reviewers,

Our sincere thanks for the in depth feedback and comments provided for this work, they helped a lot in the revisions we made. We also hope the rebuttal resolves your questions and concerns. As we're approaching the end of the discussion period, please don't hesitate to post any further questions / comments you have.

Thank you.

---

### Meta-Review · Area_Chair_JaSn · 2023-12-05

**Metareview:**

This submission, focusing on integrating functional differentiation into JAX, has received varying assessments from the reviewers. The paper presents a new package for JAX that introduces functional differentiation. The reviewers appreciated the strong theoretical foundation and the practical applications demonstrated. However, the reviewers also noted some weaknesses in term of clarity. To enhance its impact, the authors should address the reviewers' concerns, particularly by clarifying the theoretical results.

**Justification For Why Not Higher Score:**

Lack of clarity / explanation in the current version, and no higher-order version

**Justification For Why Not Lower Score:**

Strong foundation and interesting ideas.

---

### Decision · Program_Chairs · 2024-01-16

Accept (poster)